# Cell competition corrects noisy Wnt morphogen gradients to achieve robust patterning in the zebrafish embryo

Yuki Akieda [1,2], Shohei Ogamino [1], Hironobu Furuie[3,4], Shizuka Ishitani[1], Ryutaro Akiyoshi[5], Jumpei Nogami[6], Takamasa Masuda[3], Nobuyuki Shimizu[3], Yasuyuki Ohkawa[6] & Tohru Ishitani [1,2,3]*

Morphogen signalling forms an activity gradient and instructs cell identities in a signalling strength-dependent manner to pattern developing tissues. However, developing tissues also undergo dynamic morphogenesis, which may produce cells with unfit morphogen signalling and consequent noisy morphogen gradients. Here we show that a cell competition-related system corrects such noisy morphogen gradients. Zebrafish imaging analyses of the Wnt/β-catenin signalling gradient, which acts as a morphogen to establish embryonic anterior-posterior patterning, identify that unfit cells with abnormal Wnt/β-catenin activity spontaneously appear and produce noise in the gradient. Communication between unfit and neighbouring fit cells via cadherin proteins stimulates apoptosis of the unfit cells by activating Smad signalling and reactive oxygen species production. This unfit cell elimination is required for proper Wnt/β-catenin gradient formation and consequent anterior-posterior patterning. Because this gradient controls patterning not only in the embryo but also in adult tissues, this system may support tissue robustness and disease prevention.

[1] Laboratory of Integrated Signaling Systems, Department of Molecular Medicine, Institute for Molecular & Cellular Regulation, Gunma University, Gunma 371-8512, Japan. [2] Department of Homeostatic Regulation, Division of Cellular and Molecular Biology, Research Institute for Microbial Diseases, Osaka University, Osaka 565-0871, Japan. [3] Medical Institute of Bioregulation, Kyushu University, Fukuoka 812-8582, Japan. [4] Graduate School of Medical Sciences, Kyushu University, Fukuoka 812-8582, Japan. [5] Biological Evaluation Technology 2, Research and Development, Olympus Corp., Tokyo 192-8512, Japan. [6] Division of Transcriptomics, Medical Institute of Bioregulation, Kyushu University, Fukuoka 812-8582, Japan. *email: ishitani@biken.osaka-u.ac.jp

Tissue patterning is a fundamental process during embryonic development and adult tissue homeostasis. To reproducibly achieve precise tissue patterning, the molecular and cellular systems controlling patterning must be robust against environmental and physiological perturbations. Activity gradients of morphogen signalling, including Wnt/β-catenin, bone morphogenetic protein (BMP), sonic hedgehog (Shh), fibroblast growth factor (FGF) and nodal signalling pattern the tissue axes[1,2]. In the embryonic anterior–posterior (AP) axis formation of deuterostomes (amphioxus, fish, frog and mammal), Wnt/β-catenin signalling is activated and inhibited in the presumptive posterior and anterior tissue, respectively. This bi-directional regulation forms a signalling activity gradient along the AP axis to establish embryonic AP patterning[3]. However, rapid cell proliferation and movement in developing tissues may affect morphogen diffusion and signal transduction, thereby producing cells with unfit signalling and consequent noisy morphogen gradients. It is not yet completely understood how these noises are overcome to generate robust patterning.

Cell competition is an interactive process wherein cells compete for fitness in a tissue environment. Specifically, relatively higher fitness cells eliminate those with lower fitness[4,5]. Key features of this process include that it originates from specific interactions between two cell types leading to elimination in one, and context-dependency. For example, mosaically introduced polarity-deficient cells are apoptotically eliminated by their neighbouring wild-type cells in Drosophila imaginal disc and mammalian cultured cells[6–8] and in Myc-low-level cells upon communicating with Myc-high-level cells[9–11]. Although cell competition, which is evolutionarily conserved from insects to mammals, may assist in proper embryogenesis, tissue morphogenesis, and tumour progression and prevention[12], its physiological relevance and detailed mechanisms, especially of unfit cell-sensing, remain unclear.

Here, we identify a cell competition-related system for correcting the noise in the Wnt/β-catenin morphogen gradient, presenting a previously unidentified physiological role of cell competition and the mechanisms that mediate unfit cell sensing and elimination.

## Results

**Unfit cell elimination smoothens the Wnt/β-catenin gradient.** To clarify the entire morphogen gradient formation process, we visualized Wnt/β-catenin signalling activity during AP axis formation in zebrafish early embryos (Fig. 1a) using OTM (Optimal TCF Motif):d2EGFP[13] and OTM:ELuc-CP (Supplementary Fig. 1a) reporters. These respectively express destabilized EGFP (d2EGFP), providing high spatial resolution, and highly-destabilized Emerald luciferase (ELuc-CP), possessing high temporal resolution and suitable for quantitative analyses (Supplementary Fig. 1b–e), upon Wnt/β-catenin signalling activation. A noisy signalling-gradient along the AP axis was detected in both transgenic zebrafish embryo types at around 8.5–12 h-post-fertilization (hpf) (Fig. 1b–d, Supplementary Movie 1). Abnormally low and high Wnt/β-catenin activities were spontaneously detected in the Wnt/β-catenin activity-high posterior and -low regions, respectively (Fig. 1b, d, e, Supplementary Movie 1). We confirmed that the endogenous Wnt/β-catenin target gene (lef1) and nuclear β-catenin proteins also showed noisy expression patterns, which was reflected by the reporter activities (Supplementary Fig. 1f–i). Abnormal Wnt/β-catenin activity gradually disappeared over time (Fig. 1d), suggesting that zebrafish embryonic tissue may possess a system for eliminating signalling noise to smoothen the Wnt/β-catenin-gradient. As mouse embryonic tissues eliminate defective cells, including low Myc

level, autophagy-deficient, and tetraploid cells in an apoptosis-dependent manner[9,11], unfit Wnt/β-catenin activity-abnormal cells might also be apoptotically eliminated. To investigate the relationship between apoptosis and abnormal Wnt/β-catenin activity, we detected active caspase-3- and TUNEL-positive apoptotic cells in 8–10 hpf embryos undergoing Wnt/β-catenin-gradient-mediated AP axis patterning (Supplementary Fig. 2a–c, Supplementary Movie 2). Apoptotic cell number and position varied between embryos (Supplementary Fig. 2d and 2e), suggesting that the apoptosis is not pre-programmed. In some unfit Wnt/β-catenin activity-abnormal cells, caspase-3 was activated (Fig. 1b, right; Supplementary Fig. 2f–g), whereas apoptosis inhibition by anti-apoptotic bcl-2 or caspase inhibitor p35 overexpression reduced physiologically occurring apoptosis (Supplementary Fig. 2c–h), enhanced the appearance of unfit cells with abnormally high or low Wnt/β-catenin activity, and severely distorted the Wnt/β-catenin activity gradient (Fig. 1f, Supplementary Fig. 2i–j). These results suggest that apoptotic elimination of unfit cells smoothens the Wnt/β-catenin-gradient.

**A large β-catenin activity differential triggers apoptosis.** To confirm that embryonic tissue equips the system for eliminating cells with unfit Wnt/β-catenin activity via apoptosis, we artificially introduced a small number of GFP-expressing Wnt/β-catenin-abnormal cells into zebrafish embryos by injecting heat-shock-driven expression plasmids (Fig. 2a) and tracking their behaviour. As predicted, Wnt/β-catenin-hyperactivated cells expressing constitutively active β-catenin (β-catCA), a Wnt receptor LRP6 (LRP6CA), or a dominant-negative Wnt/β-catenin negative regulator GSK-3β mutant (GSK-3β DN) activated caspase-3 and gradually disappeared, whereas negative control cells expressing GFP alone survived. GSK3βDN can mimic physiological Wnt/β-catenin activation in zebrafish embryos (Supplementary Fig. 2k). Wnt/β-catenin-inactivated cells expressing Wnt/β-catenin negative regulators Axin1 or GSK-3β or a dominant-negative LRP6 mutant also activated caspase-3 and were eliminated (Fig. 2b–d, Supplementary Movies 3 and 4). The Wnt/β-catenin-hyper-activated or -inactivated cells underwent DNA fragmentation (Supplementary Fig. 3a) and were TUNEL-positive (Supplementary Fig. 3b). Caspase-inhibitor p35 co-expression blocked β-catCA-expressing cell elimination from larval tissue (Fig. 2d). These results suggest that the artificially introduced Wnt/β-catenin-abnormal cells are apoptotically eliminated.

Zebrafish embryos consist of epithelial (envelope layer) and mesenchymal tissues (deep cells)[14]. In both tissues, Wnt/β-catenin-hyperactivated cells activated caspase-3 and then disappeared (Supplementary Fig. 3c and 3d), indicating that both tissues eliminate Wnt/β-catenin-abnormal cells. Mosaic but not ubiquitous introduction of abnormal Wnt/β-catenin activation or inhibition into normal embryos strongly activated caspase-3 (Fig. 2e). Mosaically introduced β-catCA-expressing cells were efficiently eliminated from normal embryos but not from Wnt/β-catenin-hyperactivated embryos injected with antisense morpholino (MO) against the Wnt/β-catenin negative regulator APC or treated with the GSK-3β chemical inhibitor BIO (Fig. 2f and Supplementary Fig. 3d). Small numbers of β-catCA-expressing cells transplanted into normal but not β-catCA-expressing embryonic tissue activated caspase-3 in cells contacting normal cells (Supplementary Fig. 3e). These results indicate that Wnt/β-catenin activation or inhibition is not sufficient to induce apoptosis and suggest that Wnt/β-catenin-abnormal cells require surrounding healthy cells with appropriate Wnt/β-catenin activity for apoptosis induction. The Wnt/β-catenin activity difference between the abnormal and neighbouring cells thus appears to trigger apoptosis in the former. Accordingly, cells with

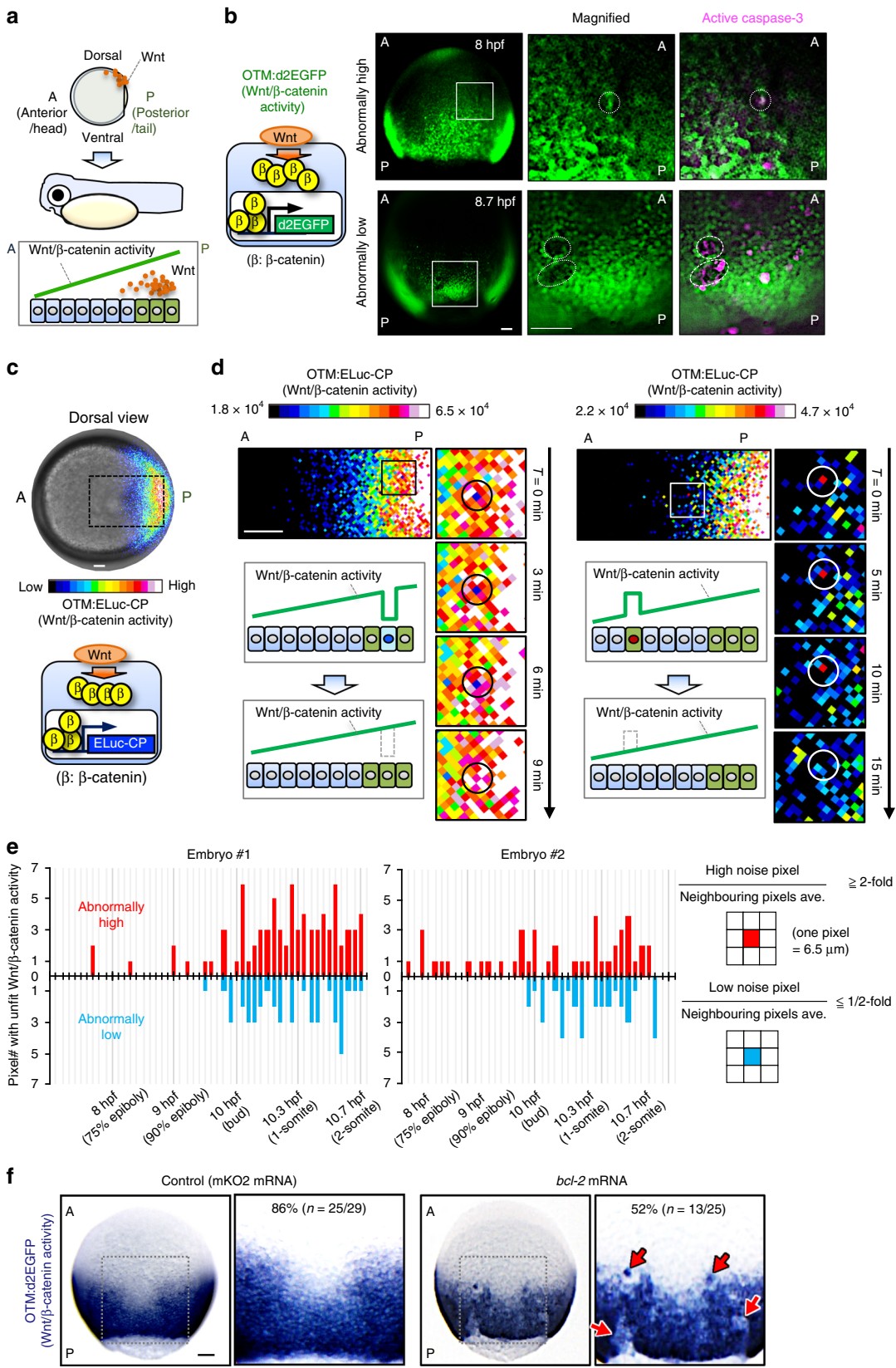

abnormally high or low Wnt/β-catenin activity efficiently activated caspase-3 in the Wnt/β-catenin signalling-low anterior or signalling-high posterior region, respectively (Fig. 2g; Supplementary Fig. 3f). This result indicates that a large Wnt/β-catenin activity differential between abnormal and neighbouring cells is required for inducing abnormal cell apoptosis. Mosaic activation of neither Ras nor Src signalling could activate caspase-3 in zebrafish embryos (Supplementary Fig. 3g), indicating that zebrafish embryonic tissue specifically kills Wnt/β-catenin-abnormal cells. However, survival of keratinocytes with

**Fig. 1** Apoptotic elimination of unfit cells smoothens the Wnt/β-catenin gradient. **a** Schematic illustration of Wnt/β-catenin activity gradient formation. A: anterior, P: posterior. **b** Caspase-3 activation in unfit cells with abnormal Wnt/β-catenin activity. Whole-mount immunostaining of d2EGFP (green) and active caspase-3 (magenta) in Tg(OTM:d2EGFP) zebrafish embryos (Dorsal view). Dotted line indicates abnormal Wnt/β-catenin-reporter activity. Scale bars, 200 μm. **c** OTM:ELuc-CP drives destabilized ELuc-CP expression in response to Wnt/β-catenin signalling activation in reporter embryo (dorsal view). Scale bar, 200 μm. (See also Supplementary Movie 1). **d** Time lapse images showing unfit cells with abnormal Wnt/β-catenin activity appear then disappear in OTM:ELuc-CP embryos. Scale bars, 100 μm. Pixel area length is 6.5 μm, ≤ zebrafish deep cell diameter (~10 μm). **e** Physiological Wnt/β-catenin-noise during zebrafish AP axis formation. Graphs show the number of pixels with unfit Wnt/β-catenin activity in the luminescence images of living OTM:ELuc-CP transgenic zebrafish embryos during AP axis formation. Schematic illustrations: pixel retaining >two-fold or <two-fold intensity compared to neighbouring pixels for ≥frames (>6 min) was defined as High or Low noise, respectively. Pixels retained for ≥two frames were counted as the physiological Wnt/β-catenin noise. Pixels spontaneously showing abnormally high or low activity within one frame were regarded as other noise (e.g., cosmic rays and detector noises) and excluded. **f** Apoptosis inhibition enhances unfit abnormally high or low Wnt/β-catenin activity cell appearance. Whole-mount in situ hybridization of d2EGFP in Tg(OTM:d2EGFP) embryos (dorsal view). $p < 0.01$ (Fisher's exact test). Scale bar, 200 μm. See also Supplementary Figs. 1 and 2

abnormally high and low Wnt/β-catenin activity in zebrafish larval skin (Supplementary Fig. 3h–i) suggests that the Wnt/β-catenin-abnormal cell elimination system may specifically exist in tissues forming Wnt/β-catenin-gradients.

**Cadherin proteins form gradients along the AP axis**. Next, we investigated the mechanisms for sensing cells with unfit Wnt/β-catenin activity. Mosaic introduction of constitutive active or dominant negative β-catenin nuclear effector Lef1 (Lef1CA and Lef1DN) mutants did not activate caspase-3 (Fig. 3a, b). Lef1DN co-expression did not block caspase-3 activation in β-catCA-expressing cells (Fig. 3a, b), suggesting that unfit Wnt/β-catenin signalling triggers apoptosis independent of Lef1. Accordingly, β-catCA proteins localized to the nucleus, cytoplasm, and membrane (Fig. 3c), whereas mosaic nuclear-export-signal (NES)-tagged β-catCA expression, which mainly localised to the cytoplasm and membrane (Fig. 3c), activated caspase-3 (Fig. 3a, b). These results suggest that unfit Wnt/β-catenin signalling in the cytoplasm or membrane, but not in the nucleus, triggers apoptosis.

Because β-catenin binds to the adhesion molecule cadherin in the membrane[15], we hypothesized that cadherin may facilitate unfit cell apoptosis. Mosaic introduction of mutant β-catCA Y654E, with transactivation but not cadherin-binding activity[16–18], could not activate caspase-3 (Fig. 3a, b). This result suggests that direct β-catenin-cadherin interaction may be required to induce unfit cell apoptosis. Immunostaining of membrane β-catenin and cadherin expression in zebrafish embryos detected Wnt/β-catenin activity (OTM:d2EGFP) gradient formation and nuclear β-catenin level as well as membrane β-catenin and cadherin gradient formation along the AP axis (Fig. 3d). Cells in the Wnt/β-catenin-high posterior region expressed high nuclear and membrane β-catenin and cadherin levels, albeit relatively low levels in the Wnt/β-catenin-low anterior region (Fig. 3d), suggesting that Wnt/β-catenin signalling may promote membrane β-catenin and cadherin accumulation. Accordingly, Wnt antagonist Dkk1 overexpression in whole embryos reduced membrane β-catenin (Fig. 3e) and E-cadherin protein levels (Fig. 3f), whereas β-catenin overexpression increased E-cadherin protein levels (Fig. 3f). Conversely, Dkk1 or β-catenin overexpression did not affect E-cadherin mRNA levels (Fig. 3g), suggesting that Wnt/β-catenin signalling post-translationally stabilises E-cadherin in zebrafish embryos.

**Cadherin is involved in unfit cell sensing**. As expected, upon introducing abnormally high Wnt/β-catenin activity into the cadherin level-low anterior tissue, Wnt/β-catenin-abnormally high cells increased both endogenous and exogenous cadherin proteins, with the converse also being true (Fig. 4a, b, and Supplementary Fig. 4a–c). Similar endogenous cadherin-level

change was also detected in naturally generated unfit cells (Supplementary Fig. 4d). These results suggest that unfit cells alter cadherin levels to yield substantial differences of membrane cadherin levels (cadherin imbalance) between unfit and neighbouring normal cells. We hypothesized that this imbalance may be involved in unfit cell elimination. To test this, we relieved the cadherin imbalance by partial knockdown (Supplementary Fig. 4e) or E-cadherin overexpression in whole embryos. Both treatments reduced Wnt/β-catenin abnormally high- and low- cell apoptosis (Fig. 4c, d). Mosaic introduction of cells overexpressing E-cadherin was sufficient to induce caspase-3 activation (Fig. 4e). To avoid β-catenin absorption (and consequent Wnt/β-catenin activity reduction) by overexpressing E-cadherin, we also examined the E-cadherin αC mutant lacking β-catenin-binding activity. E-cadherin αC mutant-expressing cells also activated caspase-3 (Fig. 4e), suggesting that abnormally high levels of cadherin stimulate apoptosis downstream of unfit β-catenin signalling. In addition, small numbers of E-cadherin-partial knockdown cells transplanted into normal embryonic tissue efficiently activated caspase-3 in the posterior tissue, which expresses high levels of cadherin (Fig. 4f and Supplementary Fig. 4f), suggesting that abnormally low levels of cadherin also stimulate apoptosis. Thus, the membrane cadherin level difference between unfit and neighbouring fit cells appears to be involved in unfit cell sensing.

**TGF-β-Smad signalling mediates unfit cell killing**. We next explored the unfit cell-killing system. As previous studies suggest a role for JNK MAPK, p38 MAPK, or p53 in cell competition-mediated unfit cell apoptosis[8,19,20], we tested their potential involvement in Wnt/β-catenin-abnormal cell apoptosis in zebrafish embryos. β-catCA-expressing cells activated caspase-3 regardless of dominant negative JNK or p38 mutant expression (Supplementary Fig. 5a, b), treatment with chemical inhibitors against JNK and p38, or MO-mediated p53 knockdown (Supplementary Fig. 5c, d). Thus, JNK, p38, and p53 appear not to mediate unfit-Wnt/β-catenin-activity cell apoptosis.

To identify the actual mediators, we screened the genes involved in unfit cell-killing. We collected β-catCA-expressing cells from β-catCA-mosaically introduced (Mosaic) or β-catCA ubiquitously-expressing (Ubiquitous) embryos, or cells from uninjected embryos (Uninjected) by using FACS (Fig. 5a), and compared gene expression patterns using RNA-Seq (Fig. 5a and Supplementary Fig. 5e). Up- or downregulated genes in 'Mosaic' but not in 'Ubiquitous', compared with 'Uninjected' cells should be due to unfit cell introduction rather than simple Wnt/β-catenin activation. Notably, a negative regulator of Smad signalling, *skilb*, was downregulated in 'Mosaic' cells, whereas several Smad-target genes[21] including *lnx1*, *foxo3b*, *sesn3*, *h2afy2*, *ccng1* and *pkdccb*, were upregulated in unfit cells (Fig. 5a and

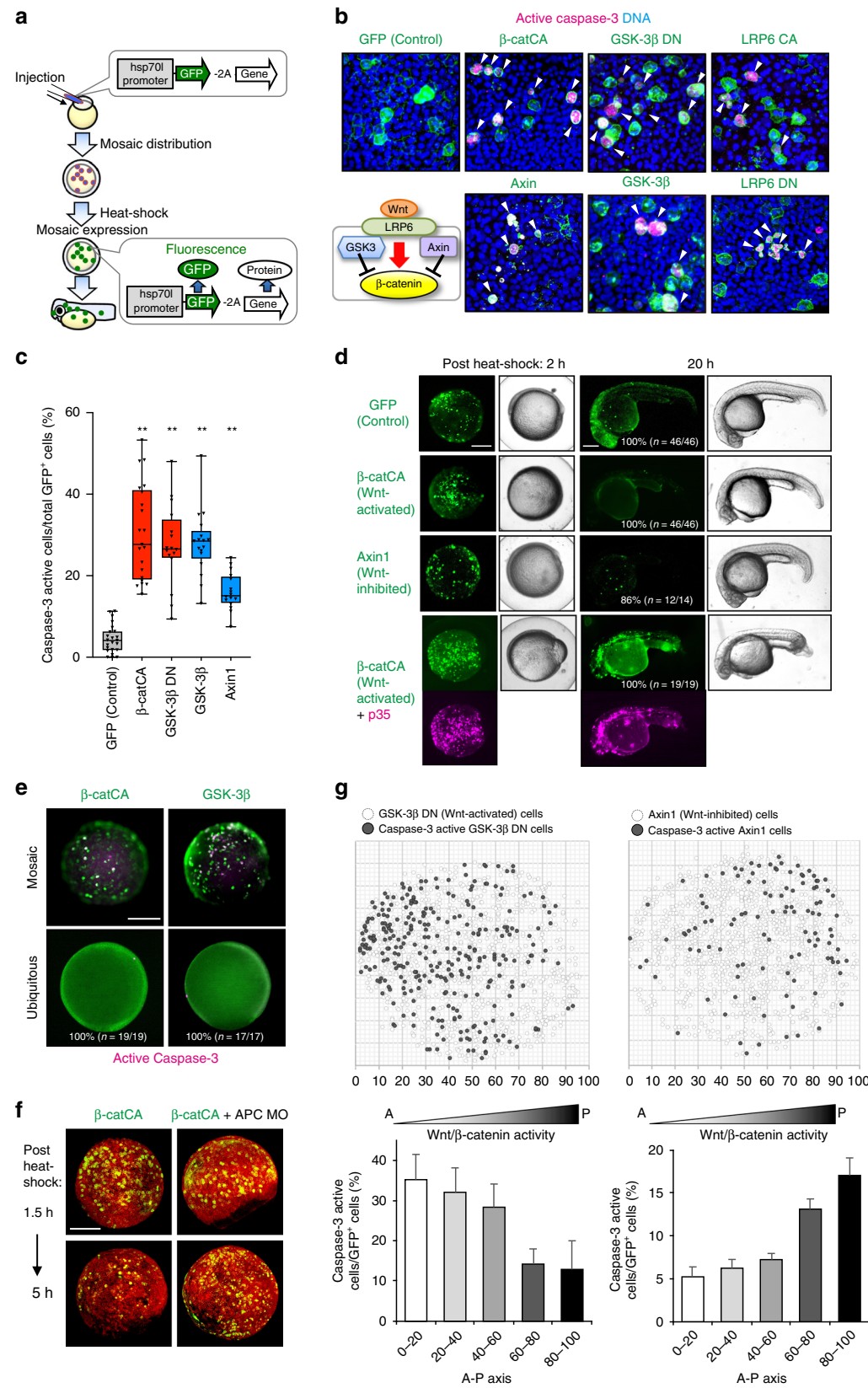

Supplementary Fig. 5e, check-marks). We also confirmed that *skilb* expression was downregulated in naturally generated unfit cells with abnormally high or low Wnt/β-catenin activity (Supplementary Fig. 5f). These results suggest that Smad signalling is upregulated in unfit cells. Smad family proteins

consist of R-Smad (Smad1/2/3/5/8), co-Smad (Smad4) and I-Smad (Smad6/7). Upon stimulation with TGF-β superfamily of secreted ligands including TGF-βs and BMPs, R-Smad is phosphorylated and forms a complex with Smad4, which translocates into the nucleus to activate target gene expression.

**Fig. 2** Substantial difference in Wnt/β-catenin activity between unfit and neighbouring cells triggers unfit cell apoptosis. **a** Schematic illustration of experimental introduction of fluorescent Wnt/β-catenin-abnormal cells in zebrafish early embryo through heat shock induction. **b**, **c** Artificially introduced Wnt/β-catenin-abnormal cells undergo apoptosis. Confocal microscopy images showing whole-mount immunostaining of active caspase-3 (magenta) in mosaic embryos expressing membrane GFP ± Wnt activators or inhibitors. Scale bar, 50 μm. Arrowheads indicate caspase-3-active cells. **c** Box plots of GFP$^+$ caspase-3-active cell frequencies show 75th, 50th (median), and 25th percentiles (right). Whiskers indicate minimum and maximum. Each dot represents one embryo ($n = 23, 21, 16, 17$ and 16 embryos, two or more independent experiments). $**p < 0.01$ (one-way ANOVA). **d** Embryos artificially introduced with cells expressing membrane GFP alone (GFP) or with β-catCA or Axin1 with or without caspase inhibitor p35. Fluorescence and bright-field images after heat-shock. p35 expression (magenta). Scale bar, 200 μm. (See also Supplementary Movies 3 and 4). **e** Surrounding normal cells are required for apoptosis induction of β-catCA- or GSK-3β-overexpressing cells. [Tg(hsp70l:GFP-T2A-β-catCA)] or [Tg(hsp70l:GFP-T2A-GSK-3β)] transgenic lines were exposed to heat shock. Percentages of embryos showing similar phenotype and number of embryos are shown. Scale bar, 200 μm. **f** Alleviation of Wnt/β-catenin activity difference between β-catCA-overexpressing cells and surrounding normal cells by injecting APC MO blocks β-catCA-overexpressing cell elimination. Cell membrane was visualized by injecting membrane-tagged mKO2 mRNA (red). Scale bar, 200 μm. **g** Cells causing excess noise in Wnt/β-catenin-gradients efficiently undergo apoptosis. Top panels show the maps of artificially introduced cells in zebrafish embryos. Bottom graphs show the means ± SEM of caspase-3-active cell frequencies within the divided range along the AP axis (GSK-3βDN, $n = 8$ embryos, 1131 cells; Axin1, $n = 14$ embryos, 1308 cells). See also Supplementary Fig. 3

In general, Smad2/3 mediate TGF-β signalling, whereas Smad1/5/8 mediates BMP signalling[22]. A TGF-β-type Smad, Smad2, translocated into the nucleus in both artificially introduced and naturally generated unfit cells with abnormally high and low Wnt/β-catenin activity (Fig. 5b and Supplementary Fig. 5g). SBE-luc, which expresses luciferase in response to TGF-β-type Smads-dependent signalling pathway activation, was also activated in both abnormal cells (Fig. 5c). In contrast, Smad2 nuclear translocation and SBE-luc activation were not detected in AP-axis forming zebrafish embryos (Fig. 5b, c). Smad1/5/8 phosphorylation, which is activated in zebrafish presumptive ventral tissue (Supplementary Fig. 5h) to promote dorsoventral axis formation[23], was not detected in Wnt/β-catenin unfit cells (Supplementary Fig. 5i). These results indicate that the unfit cells activate TGF-β-Smad signalling, which is not activated in normal embryonic tissue, but not BMP-Smad signalling, which controls embryonic dorsoventral patterning in parallel with Wnt/β-catenin signalling.

We next examined the involvement of TGF-β-Smad signalling in unfit cell-killing. TGF-β-Smad signalling inhibition by *smad4* MO (Fig. 5d), *smad2/3* DN mutant mRNA (Fig. 5e), or *skilb* mRNA (Supplementary Fig. 5j) injection reduced apoptosis of both Wnt/β-catenin-abnormally high and low cells. Mosaic introduction of cells expressing a constitutively active Smad3 mutant (Smad3CA) activated caspase-3 (Fig. 5f). Mosaic introduction of the E-cadherin αC mutant activated the nuclear translocation of Smad2 (Fig. 5b) and SBE-luc (Fig. 5c) whereas *smad2/3* DN mutants or *skilb* overexpression blocked E-cadherin αC mutant-induced caspase-3 activation (Fig. 5g). These results suggest that TGF-β-Smad signalling mediates unfit cell-killing downstream of cadherin.

**ROS kill unfit cells downstream of cadherin and Smad.** Smad signalling promotes ROS production[24]. *foxo3b* and *sesn3*, Smad-target genes upregulated in unfit cells (Fig. 5a and Supplementary Fig. 5e), are also involved in ROS-mediated apoptosis[25]. RNA-Seq data also showed that negative regulators of ROS (*aplnrb*, *hsp70.3*, *sephs1* and *dkc1*)[26–29] were downregulated in unfit cells, whereas positive regulators (*fabp3*, *ptgdsb.1* and *krt8*)[30–32] were upregulated (Fig. 5a and Supplementary Fig. 5e). Sephs1 expression was downregulated in both Wnt/β-catenin abnormally high and low cells, as determined by immunostaining (Fig. 6a). ROS may thus be activated in unfit cells. Accordingly, mosaic introduction of abnormal Wnt/β-catenin activation or inhibition into normal embryos activated ROS production and DNA oxidization (Fig. 6b, c), whereas ubiquitous Wnt/β-catenin activation or inhibition did not activate ROS production (Supplementary Fig. 6). Co-expression of the ROS negative regulators SOD1[33] or

Sephs1[29], or *smad4* knockdown reduced ROS and DNA-oxidization levels in β-catCA-expressing cells (Fig. 6b, c). ROS production and DNA oxidization were also induced in cells expressing Smad3CA or the E-cadherin αC mutant (Fig. 6b, c). These results suggest that cells with unfit Wnt/β-catenin activity activate ROS production through cadherin and Smad signalling. Furthermore, SOD1 or Sephs1 overexpression blocked caspase-3 activation in the artificially introduced Wnt/β-catenin abnormally high and low cells (Fig. 6d), suggesting that ROS mediates apoptosis of unfit cells. Previous studies reported that ROS induces apoptosis by reducing the levels of an anti-apoptotic Bcl-2 protein[34,35], raising the possibility that Bcl-2 reduction may be involved in ROS-mediated apoptosis of unfit cells. We found that levels of exogenous Bcl-2 protein in Wnt/β-catenin-high or -low unfit cells were significantly lower than those in normal control cells (Fig. 6e), and that overexpression of SOD1 rescued the Bcl-2 reduction in Wnt/β-catenin-high unfit cells (Fig. 6e), suggesting that ROS-mediated apoptosis of unfit cells is likely to be trigger through the reduction of Bcl-2 protein levels.

**Unfit cell elimination is required for precise patterning**. To confirm the importance of elimination of Wnt/β-catenin-abnormal cells for embryogenesis, we examined the effects of inhibiting unfit cell apoptosis. Blocking apoptosis by p35 co-expression in cells with artificially introduced β-catCA induced ectopic expression and abnormal reduction of brain AP markers (Supplementary Fig. 7a). Co-expression of p35 with β-catCA or GSK-3β increased abnormal morphogenesis (Supplementary Fig. 7b), whereas expression of p35, β-catCA, or GSK-3β alone had little effect.

To further test the physiological significance of eliminating unfit cells, we blocked ROS-mediated apoptosis of unfit cells by overexpressing a ROS negative regulator. Sephs1 overexpression partially reduced physiologically occurring apoptosis (Fig. 7a). SOD1 or Sephs1 overexpression induced both ectopic activation and abnormal reduction of the Wnt/β-catenin reporter (Fig. 7b), endogenous *lef1* mRNA expression (Supplementary Fig. 7c), and brain AP markers (Fig. 7c, Supplementary Fig. 7d) and disturbed embryonic morphogenesis (Fig. 7d), as did treatment with the ROS scavenger *N*-acetyl-ʟ-cysteine (NAC) (Fig. 7b). These data suggest that ROS-mediated elimination of naturally generated Wnt/β-catenin-unfit cells is essential to achieve smooth Wnt/β-catenin-gradient formation and precise embryogenesis.

**Discussion**
The present study demonstrates a cell competition-mediated 'noise-cancelling system' for morphogen signalling-gradients

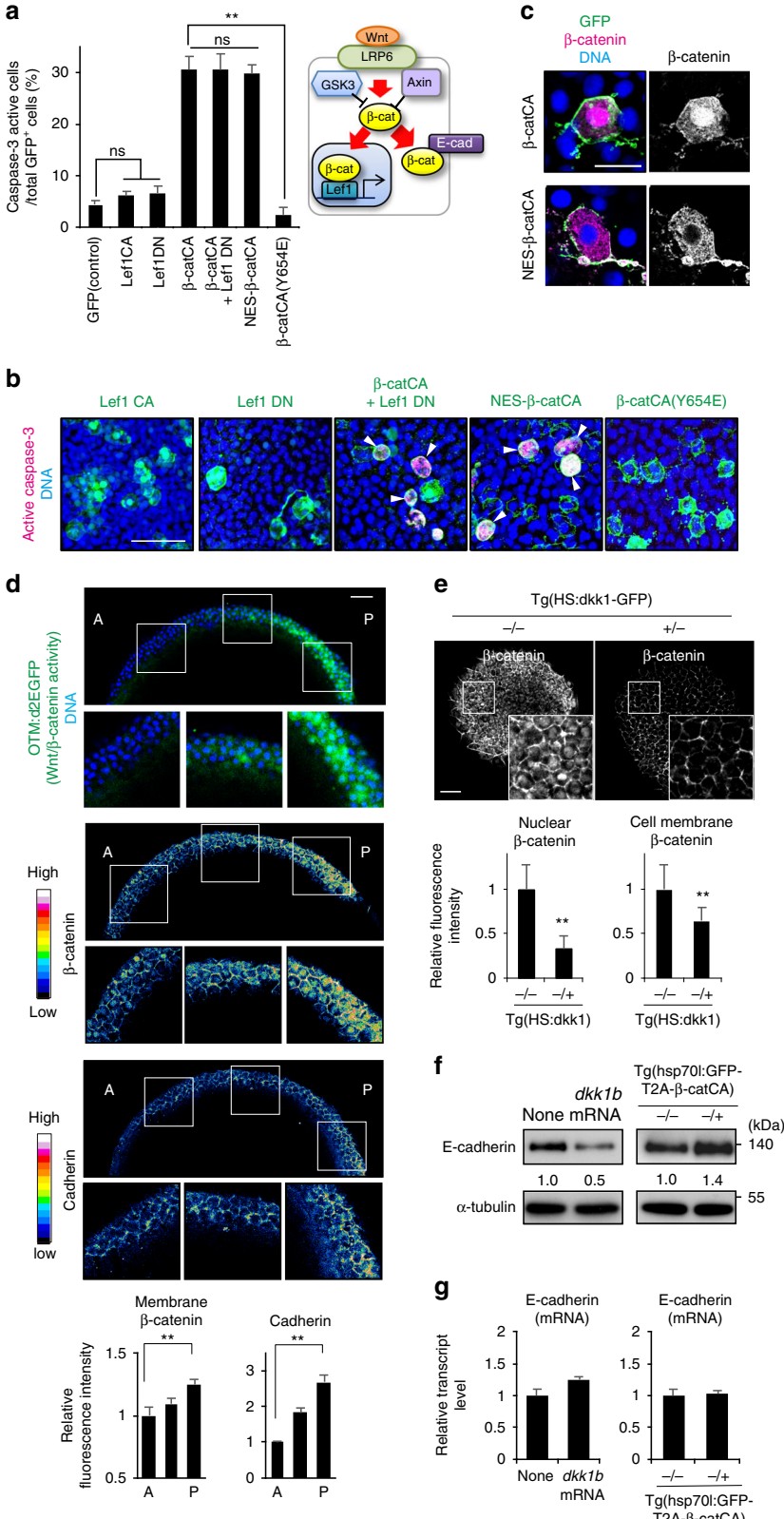

(Fig. 7e). During zebrafish embryonic AP patterning, membrane β-catenin and cadherin protein level gradients along the AP axis are formed in a Wnt/β-catenin activity-dependent manner. The spontaneous appearance of cells with unfit Wnt/β-catenin activity produces substantial noise in the Wnt/β-catenin-gradient and alters membrane β-catenin and cadherin protein levels, leading to an imbalance of cadherin levels between unfit and neighbouring cells. Unfit cells also activate TGF-β-Smad signalling to produce ROS and consequently undergo ROS-mediated apoptosis. In this manner, embryonic tissues eliminate excess noise from the Wnt/β-catenin gradient to support its proper formation along with robust embryonic patterning.

**Fig. 3** Membrane β-catenin and cadherin proteins form their concentration gradient along the AP-axis in a Wnt/β-catenin gradient-dependent manner. **a** Direct β-catenin-E-cadherin binding is required for unfit cell apoptosis induction. Schematic illustration of nuclear and membrane β-catenin functions (right). Graph show means ± SEM of GFP[+] caspase-3-active cell frequencies in Lef1 or β-catCA mutant mosaic embryos (left) (n = 9, 11, 13, 13, 31, 14 and 6 embryos, two or more independent experiments). **p < 0.01 (one-way ANOVA). **b** Representative confocal fluorescence images show GFP (green), active caspase-3 (magenta), and DNA (blue) in embryos introduced with cells expressing Lef1 mutants or β-catCA mutants. Scale bar, 50 μm. **c** Both β-catCA and NES-β-catCA localize to the plasma membrane in zebrafish embryos. Scale bar, 20 μm. **d** Membrane β-catenin and E-cadherin protein levels correlate with Wnt/β-catenin signalling activity. Optical sagittal cross-section (dorsal side) in 8.3 hpf embryos. Panels show fluorescence of OTM:d2EGFP (green) and DNA (blue), fluorescence intensity of β-catenin and E-cadherin staining, and their magnified views. Scale bar, 50 μm. Bottom graph shows fluorescence intensity (means ± SEM, n = 12 cells per region) of β-catenin and E-cadherin staining at intercellular boundaries within three evenly divided regions across the AP axis. (Also see Methods). **p < 0.01 (t test). **e** Inhibition of Wnt signalling (Dkk1 overexpression) reduces nuclear as well as membrane β-catenin. Dorsal side of whole-mount β-catenin immunostaining of Tg(HS:dkk1b-GFP) zebrafish embryos and sibling embryos at 9 hpf exposed to heat shock at 37 °C from 4.3 to 5.3 hpf. +/− and −/− indicate the heterozygous transgenic sibling and non-transgenic wild-type sibling, respectively. Scale bar, 50 μm. Bottom graph shows fluorescent intensity (means ± SEM, n = 10 cells). **p < 0.01 (t test). **f, g** E-cadherin protein level correlates with Wnt/β-catenin signalling activity. 9 hpf embryos injected with dkk1b mRNA or Tg(HS:hsp70l:GFP-T2A-β-catCA) embryos exposed heat shock at 37 °C from 4.3 to 5.3 hpf were extracted and then subjected into immunoblotting with anti-E-cadherin and anti-α-tubulin antibodies (**f**) or qPCR (**g**)

Cells artificially introduced with abnormally high or low BMP morphogen signalling activity undergo apoptosis through cell competition or its related system in *Drosophila* imaginal tissues and mouse epiblasts[11,20,36]. However, the physiological importance and detailed mechanisms of this phenomenon remain unclear. We show that both artificially and physiologically introduced unfit cells, which produce excess noise in the Wnt/β-catenin morphogen-gradient, undergo apoptosis through cadherin, TGF-β-Smad, and ROS, and identify the physiological importance of this system in signalling-gradient formation and embryonic morphogenesis. Thus, we unravelled an undescribed morphogen-gradient-correcting system that plays physiological roles in normal development. The previous findings together with our present results suggest that the apoptosis-mediated morphogen noise-cancelling system may be evolutionarily conserved, with Wnt/β-catenin and other morphogen signalling utilizing similar systems. ROS signalling inhibition partially reduced the number of physiologically appearing apoptotic cells, suggesting that these cells comprise Wnt/β-catenin-defective (ROS-activated) cells along with other cells, potentially including other morphogen signalling-defective cells. With regard to sensing 'unfitness', only one report has identified that a ligand protein Sas and its receptor PTP10D are involved in recognizing polarity-deficient cells[37]. We showed that unfit Wnt/β-catenin activity is translated to unfit cadherin levels, which are sensed by neighbouring cells with fit cadherin levels and fit Wnt/β-catenin activity.

Apoptosis-independent morphogen gradient-correcting systems also exist. In the zebrafish developing neural tube in which Shh signalling forms its activity-gradient along the dorsal-ventral axis, cells with unfit Shh signalling activity migrate to the appropriate area to form a smooth Shh signalling-gradient[38]. Morphogen signalling equips negative feedback systems mediated through negative signalling regulators such as Axin2 (in Wnt/β-catenin signalling), Smad6/7 (in BMP signalling) and patched (in Shh signalling)[39,40]. Such feedback systems also contribute to correcting morphogen signalling noise. Developing tissues may use these correction systems differently in accordance to the 'unfitness' of unfit cells. Because the apoptosis-mediated system is activated only in cells causing substantial noise, this system might contribute in particular to eliminating cells with severe signalling-defects (e.g., with mutations that constitutively activate or inhibit morphogen signalling).

Recent studies are gradually clarifying the fact that cell competition is required for proper embryogenesis and tissue morphogenesis. Inhibition of the cell competition mediator *azot* induces the accumulation of unfit cells and consequent developmental malformation in *Drosophila*[41], suggesting that cell competition acts during physiological development and assists in tissue morphogenesis. We also show that the inhibition of Wnt/β-catenin-noise cancelling system-mediated dysfunctional cell elimination disturbs proper patterning. The Wnt/β-catenin-noise cancelling system appears to comprise a kind of cell competition supporting precise tissue morphogenesis.

In developing mouse embryos, Myc-high cells compete with and kill Myc-low cells, supporting proper development[9,11]. We also observed that the mosaic introduction of cells expressing Myc at high levels killed surrounding normal cells in zebrafish embryos (Supplementary Fig. 4g). *Myc* constitutes a Wnt/β-catenin target gene, raising the possibility that cells with high Wnt/β-catenin activity might trigger Myc-driven cell competition. However, this competition was cadherin- and Smad4-independent (Supplementary Figs. 4g and 5k). In addition, overexpression of a dominant-negative mutant of Myc (Myc-DN), which lacks its transactivation domain[42,43], dramatically blocked Myc-driven cell competition (Supplementary Fig. 4h), whereas it was unable to block the elimination of Wnt/β-catenin-noise cells (Supplementary Fig. 4i). These results indicate that the Myc-mediated competition and Wnt/β-catenin-noise elimination mechanisms differ. Consistent with this idea, RNA-Seq data showed that heat shock-mediated forced activation of Wnt/β-catenin signalling for 2.5 h was insufficient to upregulate *Myc* gene expression (GEO accession code: GSE133526). In addition, a 5-h treatment with BIO, a chemical activator of Wnt/β-catenin signalling, activated the expression of *Myc* genes in zebrafish embryos, whereas 2.5-h BIO treatment was unable to activate *Myc* gene expression (Supplementary Fig. 4j), indicating that *Myc* genes serve as Wnt/β-catenin signalling-target genes in zebrafish embryos. Notably, within 2.5–3 h after the heat-shock-mediated mosaic induction of Wnt signalling regulators, we could detect Smad nuclear translocation, ROS production, and caspase activation in the noise cells, indicating that Wnt/β-catenin-noise stimulates apoptotic elimination before affecting Myc expression. Therefore, cells with unfit Wnt/β-catenin activity would die before activating Myc-driven cell competition.

Although JNK and p38 regulate apoptosis induction in many cell competition systems[8,44], they appear not to be required for the Wnt/β-catenin-noise cancelling system, which is instead mediated by cadherin-imbalance and the Smad-ROS axis. Cadherin functions in both mechanical cell-cell adhesion and cell recognition[44,45]. Cells expressing low and high cadherin levels separate out when mixed in vitro, indicating that cells can recognize quantitative differences in cadherin levels[46]. However, it remains unclear how zebrafish embryonic cells translate quantitative differences in cadherin levels (cadherin-imbalance) to 'unfitness' and consequently determine to kill unfit cells.

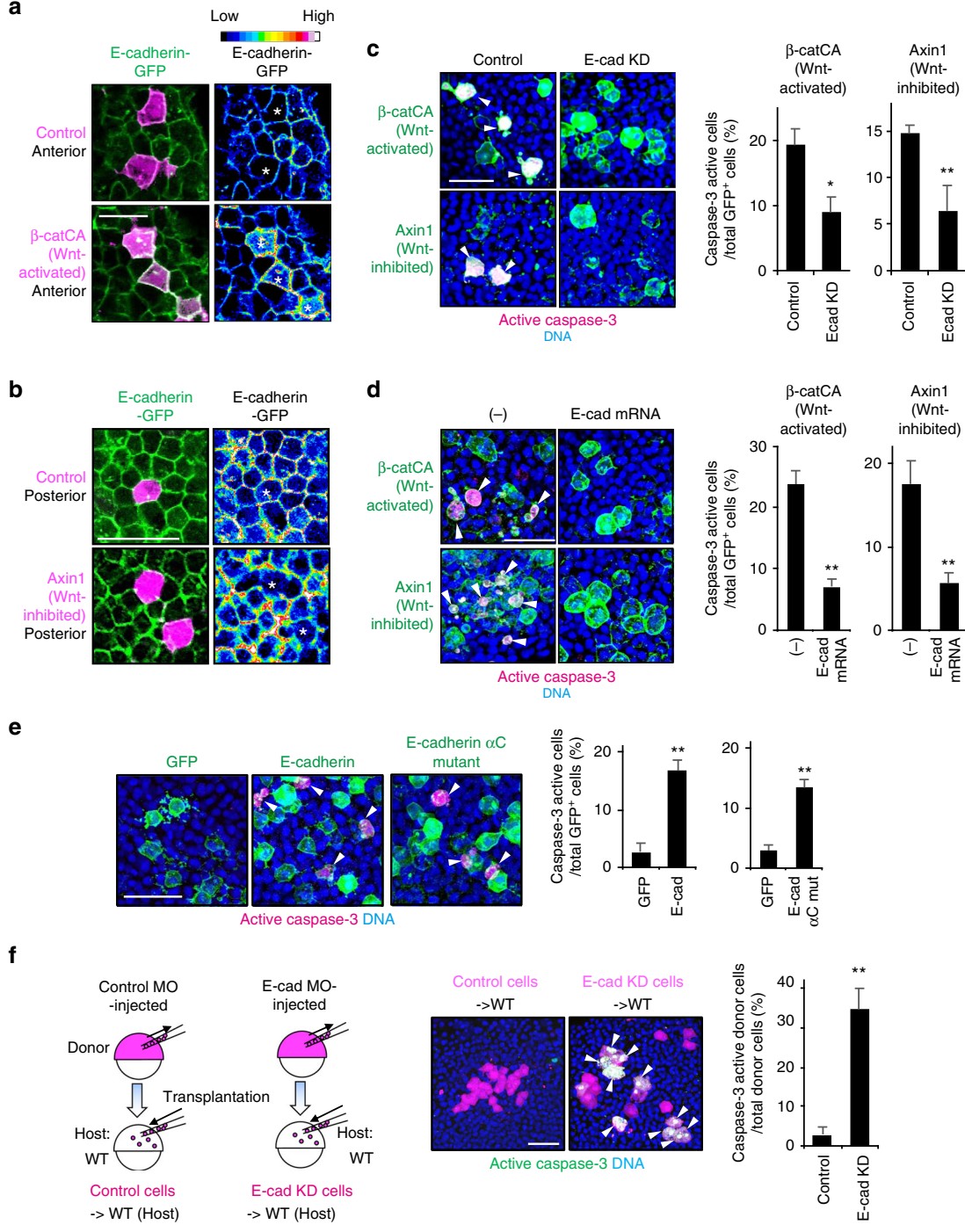

**Fig. 4** Cadherin is involved in unfit cell sensing. **a**, **b** Mosaic unfit abnormal Wnt/β-catenin-activity cell introduction changes exogenous E-cadherin levels in zebrafish embryos. Confocal images showing embryos injected with E-cadherin-GFP mRNA (green) and mosaically introduced with mKO2 alone-expressing control cells or mKO2 and β-catCA- or Axin1-overexpressing cells (magenta). Right panels show fluorescence intensity of E-cadherin-GFP. Scale bar, 50 μm. **c**, **d** Partial E-cadherin knockdown (**c**) or E-cadherin overexpression (**d**) blocks β-catCA- or Axin1-overexpressing cell apoptosis. Confocal images showing whole-mount immunostaining of active caspase-3 (magenta) in mosaic embryos expressing membrane GFP ± β-catCA or Axin1 (green) and injected with low dose E-cadherin MO (E-cad KD) or E-cadherin mRNA. Arrowheads indicate caspase-3-active cells. Scale bar, 50 μm. Graphs show the means ± SEM ($n = 8$ or more embryos, two independent experiments) of GFP+ (β-catCA, Axin1) caspase-3 active cell frequencies (right). **$p < 0.01$, *$p < 0.05$ ($t$ test). **e** Cells with unfit high E-cadherin levels undergo apoptosis in mosaic embryos. Confocal images showing whole-mount immunostaining of active caspase-3 (magenta) in mosaic embryos expressing membrane GFP ± E-cadherin or its mutant (green). Arrowheads indicate caspase-3-active cells. Scale bar, 50 μm. Graphs show the means ± SEM ($n = 5$ or more embryos, two independent experiments) of GFP+ caspase-3-active cell frequencies (right). **$p < 0.01$ ($t$ test). **f** Cells with unfit low E-cadherin levels undergo apoptosis in mosaic embryos. E-cadherin partial knockdown (E-cad KD) cells or control cells (magenta) were transplanted into wild-type embryos. Confocal images showing whole-mount immunostaining of active caspase-3 (green) in mosaic embryos. Arrowheads indicate caspase-3-active cells. Scale bar, 50 μm. Graph shows the means ± SEM ($n = 7$ or more embryos, two independent experiments) of GFP+ caspase-3-active cell frequencies (right). **$p < 0.01$ ($t$ test). See also Supplementary Fig. 4

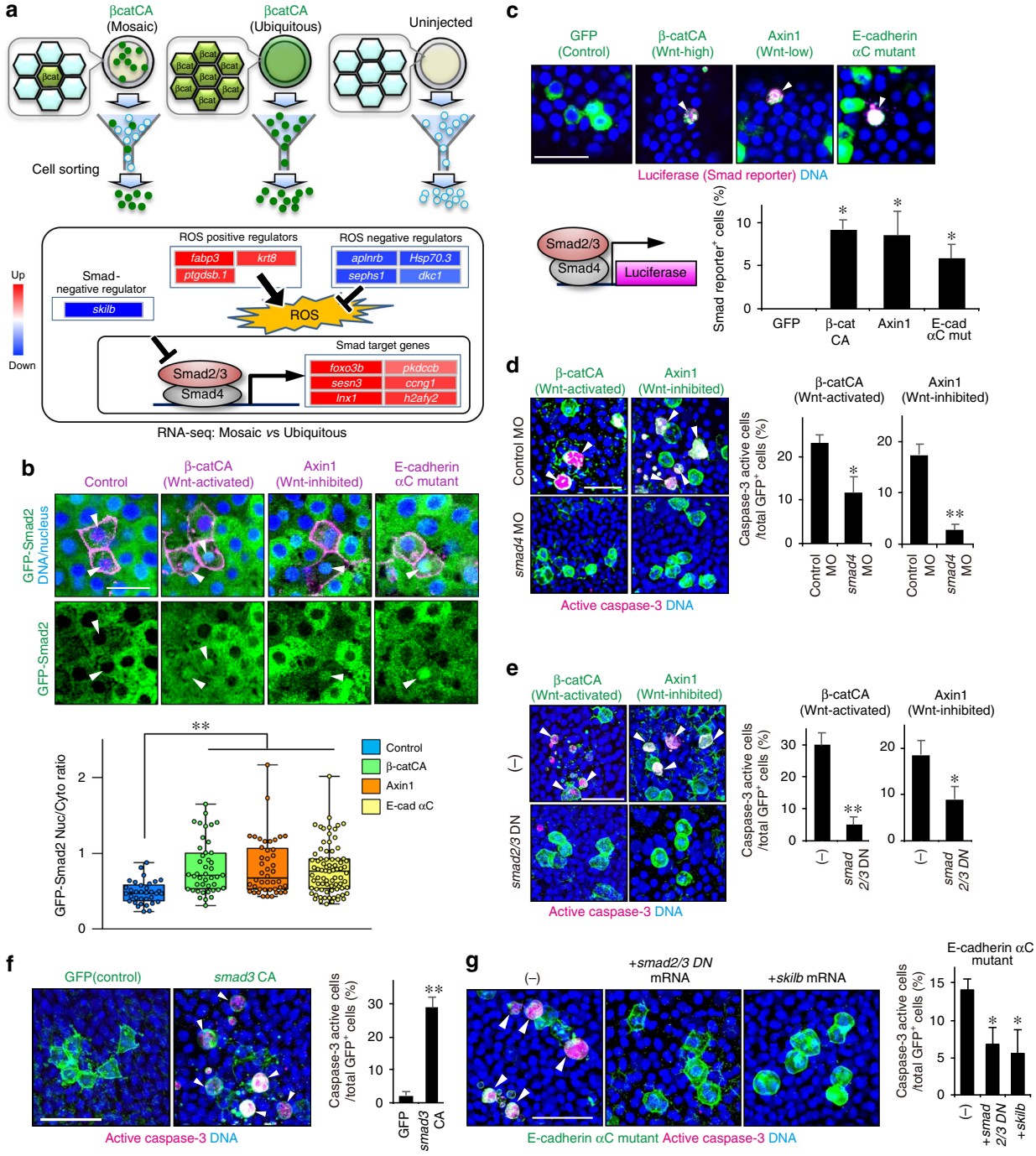

Although our results suggest that cadherin imbalance activates TGF-β-type Smads in unfit Wnt/β-catenin cells, RNA-Seq data showed that TGF-β ligands were not upregulated in unfit cells (Supplementary Fig. 5e and GEO accession code: GSE133526). In addition, treatment with a chemical inhibitor against TGF-β receptors was unable to block Smad activation and caspase activation in unfit cells (Supplementary Fig. 5l, 5m and 5n), suggesting that cadherin imbalance activates Smad in a TGF-β receptor-independent manner. Because cadherin-mediated adhesion maintains tissue integrity[45,46], local cadherin level change in unfit cells would cause local loss of tissue integrity, which may induce mechanical stress in unfit cells, in turn stimulating the unfit cell-killing system. Accordingly, mechanical stress can activate TGF-β-type Smads[47–50], which act downstream of cadherin in the Wnt/β-catenin-noise cancelling system.

However, the detailed mechanisms by which cadherin-imbalance activates TGF-β-type Smads require further investigation.

The Wnt/β-catenin-noise cancelling system may act context-dependently because cell clones with abnormally high Wnt/β-catenin activity can kill surrounding normal cells in *Drosophila* epithelial tissue (wing disc)[51], whereas Wnt/β-catenin activity-aberrant keratinocytes survive and persist in zebrafish larval skin. Clone size, cell/tissue-type, and/or intra-tissue localization might affect unfit cell behaviour. We consider that the Wnt/β-catenin-noise cancelling system acts in proliferative tissues that form a Wnt/β-catenin-gradient and kills suddenly introduced single unfit cells that produce excess noise in the gradient. Therefore, this system might also function in the intestinal crypt, which undergoes active cell turnover and also forms a Wnt/β-catenin-gradient[52]. Accordingly, in mouse intestine, mosaically introduced

**Fig. 5** TGF-β-Smad signalling mediates unfit cell killing. **a** Schematic representation of FACS (top) and mRNA expression changes by RNA-Seq between β-catCA-mosaically introduced (Mosaic), ubiquitously β-catCA-expressing (Ubiquitous), or uninjected embryos (bottom). (See also Supplementary Fig. 6e). **b** GFP-Smad2 translocates into unfit Wnt/β-catenin-activity cell nuclei. Confocal fluorescence images of cells overexpressing indicated genes in GFP-Smad2 mRNA (200 pg)-injected embryos. Scale bar, 50 μm. Box plots of nuclear/cytoplasmic GFP-Smad2 ratio show 75th, 50th (median) and 25th percentiles (right). Whiskers indicate minimum and maximum. Each dot represents one embryo. ($n = 34, 43, 44$ and $80$ cells, two independent experiments) $**p < 0.01$ (one-way ANOVA). **c** Wnt/β-catenin-abnormal cells activate a Smad2/3/4-dependent reporter gene (SBE-luc). Confocal images showing whole-mount fluorescent in situ hybridization of luciferase mRNA in mosaic embryos expressing membrane GFP ± β-catCA or Axin1 (green) and injected with SBE-luc. Arrowheads indicate SBE-luc-active cells. Scale bar, 50 μm. SBE-luc active cell frequencies are graphed. ($n = 5$ or more embryos, two independent experiments) $*p < 0.05$ (one-way ANOVA). **d, e** Smad4 knockdown (**d**) or forced expression of smad2/3 dominant negative mutants (smad2/3DN) (**e**) blocks apoptosis of βcatCA- or Axin1-expressing cells. Confocal images showing whole-mount immunostaining of active caspase-3 (magenta) in mosaic embryos expressing membrane GFP ± β-catCA or Axin1 (green) and injected with control MO, smad4 MO, or smad2/3DN mRNA. Arrowheads indicate caspase-3-active cells. Scale bar, 50 μm. Graphs show the means ± SEM ($n = 6$ or more embryos, two independent experiments) of GFP$^+$ (β-catCA- or Axin1-expressing) caspase-3-active cell frequencies (right). $**p < 0.01$, $*p < 0.05$ (t test). **f** Mosaic Smad3 activation is sufficient to induce apoptosis. Confocal images showing whole-mount immunostaining of active caspase-3 (magenta) in mosaic embryos expressing membrane GFP ± smad3CA. Arrowheads indicate caspase-3-active cells. Scale bar, 50 μm. Graph shows the means ± SEM ($n = 4$ or more embryos, two independent experiments) of GFP$^+$ caspase-3-active cell frequencies (right). $**p < 0.01$ (t test). **g** E-cadherin αC mutant-expressing cell apoptosis is inhibited in smad2/3DN- or skilb-overexpressing embryos. Confocal images showing whole-mount immunostaining of active caspase-3 (magenta) in mosaic embryos expressing membrane GFP and E-cadherin mutant and injected without (−) or with smad2/3DN or skilb mRNA. Arrowheads indicate caspase-3-active cells. Scale bar, 50 μm. Graph shows the means ± SEM ($n = 5$ or more embryos, two independent experiments) of GFP$^+$ caspase-3-active cell frequencies (right). $*p < 0.05$ (one-way ANOVA). See also Supplementary Fig. 5

Wnt signalling-hyperactivated cells strongly express E-cadherin and actively undergo apoptosis[53], indicating that the mammalian intestine may possess a similar system.

Cells with abnormally high Wnt signalling may represent an origin of cancer[54,55]. The Wnt/β-catenin-noise cancelling system may therefore also eliminate Wnt/β-catenin-hyperactivated pre-cancerous cells to prevent primary tumourigenesis. E-cadherin and Smad2/4, the mediators of this system, also act as tumour suppressors[56,57]. Their loss of function might reduce Wnt/β-catenin-noise cancelling system activity, allowing the surviving Wnt/β-catenin-high precancerous cells to prime tumourigenesis. Thus, the Wnt/β-catenin-noise cancelling system represents a facet of the roles of tumour suppressor genes. In the future, it would therefore be of interest to examine the roles of this system in cancer prevention.

## Methods

**Zebrafish maintenance**. Zebrafish were raised and maintained under standard conditions. Wild-type strains (AB and India) were used, along with and following transgenic lines: Tg(OTM:d2EGFP)[13], Tg(OTM:ELuc-CP), Tg(hsp70l:GFP-T2A-β-catCA), Tg(hsp70l:GFP-T2A-GSK-3β), and Tg(HS:dkk1b-GFP)[58]. All experimental animal care was performed in accordance with institutional and national guidelines and regulations. The study protocol was approved by the Institutional Animal Care and Use Committee of the respective universities (Kyushu University Permit# A28-005-1; Gunma University Permit# 17-051).

One-cell stage embryos were used for cell injection to generate transgenic fish or mosaic embryos, with the latter processed up to 9 hpf. One- and two cell-stage embryos were used for MO injection. Alternately, cells were injected into 3.3–3.7 hpf stage embryos.

**Mosaic Wnt/β-catenin-abnormal cell introduction**. Hsp70 promoter-driven plasmids (5–17.5 pg) were injected into one-cell-stage embryos and maintained at 28.5 °C until 4.3 hpf (dome stage). At 4.3 hpf, embryos were exposed to heat-shock. Briefly, embryos were transferred to pre-warmed egg water at 37 °C and kept at 37 °C for 1 h. After heat-shock, embryos were placed at 28.5 °C, then fixed at 9 hpf for immunostaining or in situ hybridization. This method allows introduction of the Wnt/β-catenin-abnormal cells at the single-cell level but not at the patchy-clone level; however, it is difficult to track the introduced cells for an extended duration and to strictly control the copy number of transgenes.

**Embryonic cell preparation for RNA-Seq**. β-catCA-mosaically introduced embryos were prepared as described above. HS:GFP-2A-β-catCA-transgenic zebrafish embryos were used as β-catCA-ubiquitously introduced embryos. Cell dissociation was performed using established protocol[59] with the following modification: 40 embryos per group at 8.3 hpf stage were placed in a solution of 2 mg/ml Pronase (Roche, Upper Bavaria, Germany) in E2 (15 mM NaCl, 0.5 mM KCl, 2.7 mM CaCl$_2$, 1 mM MgSO$_4$, 0.7 mM NaHCO$_3$, 0.15 mM KH$_2$PO$_4$, and 0.05 mM Na$_2$HPO$_4$) on a 2% agar-coated dish for ~6 min at 28.5 °C. After dechorionation, embryos were washed with deyolking buffer (1/2 Ginzburg Fish Ringer without calcium: 55 mM NaCl, 1.8 mM KCl, and 1.25 mM NaHCO$_3$). Embryos were transferred into a 1.5 ml tube and then the yolk was disrupted by pipetting with a 1000 μl tip. The embryos were shaken for 5 min at 2 g to dissolve the yolk (Thermomixer, Eppendorf, Hamburg, Germany). Cells were pelleted at 300 g for 1 min and the supernatant discarded. Cell pellets were resuspended in FACSmax Cell Dissociation Solution (Genlantis, San Diego, CA). β-catCA-expressing (GFP$^+$) cells were sorted by FACSAriaII (BD, Franklin Lakes, NJ). Uninjected negative control cells were also sorted under similar conditions. Sorted cells were pelleted and dissolved in TRIzol reagent (Invitrogen, Waltham, MA).

**Generation of transgenic zebrafish**. Plasmid DNA (OTM:ELuc-CP, hsp70l:GFP-T2A-β-catCA, or hsp70l:GFP-T2A-GSK-3β) along with Tol2 transposase mRNA were co-injected into one-cell stage wild-type zebrafish (AB) embryos. A transgenic fish was outcrossed with a wild-type fish to produce the founder line. To generate a transgenic line carrying a single transgene, a carrier was outcrossed with wild-type fish. Reporter transgenic fish were maintained as homozygous transgenic fish.

**Time-lapse imaging and data analysis**. For time-lapse luminescence live imaging, Tg(OTM:ELuc-CP) embryos were manually dechorionated by forceps and then mounted in 1% low melting agarose with 3 mM D-luciferin potassium salt (Wako) onto glass bottom dishes (Matsunami Glass, Osaka, Japan). Images were collected using an LV200 bioluminescence microscope (Olympus, Tokyo, Japan) equipped with a 20 × 0.75 NA objective at 28 °C. Collected image data were processed by TiLIA2 software (Olympus) for removing signals of cosmic rays, and then analysed using ImageJ with plug-ins developed by Olympus. For sequential imaging of caspase activity indicator VC3Ai fluorescence with bioluminescence, VC3Ai was excited by a halogen lamp through a 490–500 nm bandpass filter and emission was detected through 515–560 nm bandpass filter. Collected image data were processed using a convolution filter with ImageJ software.

For time-lapse confocal live imaging, embryos were manually dechorionated by forceps and mounted in 1% low melting agarose with egg water onto glass bottom dishes. Live-imaging was performed using an LSM700 confocal laser scanning microscope (Zeiss, Oberkochen, Germany) equipped with a 20 × 0.8 NA objective. Two laser lines, 488 and 559 nm were used. The recording interval was 1 to 2 min. At each time point, 12 confocal slices through the z-axis were acquired. Collected image data were processed using Imaris software (Bitplane, Zurich, Switzerland).

**Detection of ROS**. To detect ROS production, zebrafish embryos were incubated with 1.67 μM CellRox Green (Thermo Fisher) in egg water for 30 min. After loading, embryos were washed three times in egg water and fixed with 4% paraformaldehyde.

**NAC treatment**. OTM:d2EGFP-transgenic zebrafish embryos were treated with 100 μM NAC (Sigma) from the 75% epiboly to 85% epiboly stage and then fixed.

**Quantification of membrane β-catenin and E-cadherin levels**. The fluorescence intensity in two intercellular areas per cell ($n = 6$ cells) in each region was measured. Analyses were performed in duplicate.

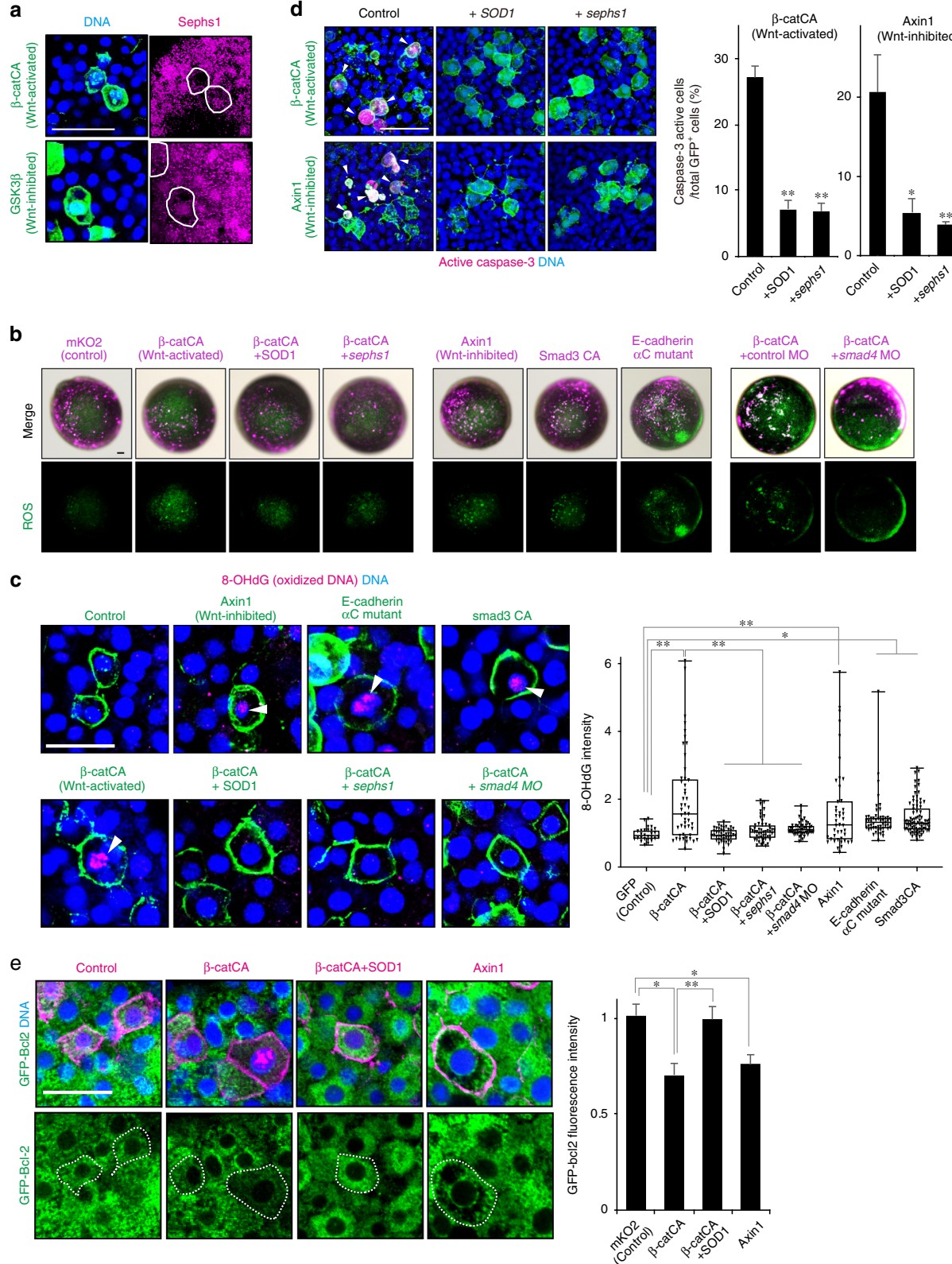

**Plasmids**. To prepare the luciferase-based reporter plasmid (OTM:ELuc-CP) (Supplementary Fig. 1a), d2EGFP of the OTM:d2EGFP plasmid was replaced with ELuc-PEST (Toyobo, Osaka, Japan)[13]. To enhance destabilization efficiency, the CL1 sequence from pGL4.28 (Promega, Madison, WI) was inserted between ELuc cDNA and the PEST sequence. To prepare heat-shock promoter-driven plasmids, the *hsp70l* promoter was sub-cloned into the pTol2 vector (a gift from Dr. K. Kawakami). Subsequently, membrane-tagged (GAP43-fused) GFP (or mKO2) and T2A were sub-cloned into pTol2-hsp70l promoter plasmids. These plasmids

express membrane GFP (or mKO2) alone in response to heat-shock. To generate the plasmids that express membrane GFP (or mKO2) with signalling modulator proteins, PCR-amplified cDNAs encoding signalling modulator proteins were sub-cloned into the downstream site of T2A of pTol2-hsp70l:GFP-T2A plasmids. Wnt/β-catenin signalling activators were as follows: N-terminus truncated mouse β-catenin (β-catCA)[60]; NES sequence (PKIα-derived)-conjugated β-catCA (NES-β-catCA); β-catCA Y654E mutant[16–18]; dominant-negative mutant of human GSK-3β (GSK-3βDN), in which Arg96 was replaced with Ala[61]; constitutively active

**Fig. 6** ROS kills unfit cells downstream of cadherin and Smad signalling. **a** Representative confocal images showing whole-mount immunostaining of Sephs1 (magenta) in mosaic embryos expressing membrane GFP ± β-catCA or GSK3β (green). Scale bar, 50 μm. **b, c** Cells with unfit Wnt/β-catenin activity activate ROS production through E-cadherin and Smad signaling. **b** Fluorescence images showing ROS probe (CellRox Green)-stained mosaic embryos expressing membrane mKO2 ± indicated genes. Scale bars, 200 μm. **c** Confocal images showing whole-mount immunostaining of 8-OHdG (magenta) in mosaic embryos expressing membrane GFP ± indicated genes (green). Arrowheads indicate 8-OHdG-positive oxidized nuclei. Scale bar, 50 μm. Box plots of fluorescence intensity of 8-OHdG staining show 75th, 50th (median) and 25th percentiles (right). Whiskers indicate minimum and maximum. Each dot represents one embryo. ($n = 40$ or more cells, two independent experiments) **$p < 0.01$, *$p < 0.05$ (one-way ANOVA). **d** ROS downregulation blocks Wnt-activated or -inhibited cell apoptosis. Confocal images showing whole-mount immunostaining of active caspase-3 (magenta) in mosaic embryos expressing membrane GFP ± β-catCA- or Axin1 (green) and injected with *SOD1* or *sephs1* mRNA. Arrowheads indicate caspase-3-active cells. Scale bar, 50 μm (left). Graphs show the means ± SEM ($n = 5$ or 6 embryos, two independent experiments) of GFP+ caspase-3-active cell frequencies (right). **$p < 0.01$, *$p < 0.05$ (one-way ANOVA). **e** Cells with unfit Wnt/β-catenin activity reduce Bcl-2 protein levels. Confocal images of mosaic embryos expressing membrane mKO2 ± β-catCA- or Axin1 (magenta) and injected with *GFP-Bcl2* mRNA. Scale bar, 50 μm (left). Graphs show the means ± SEM (n = 16 or more cells, two independent experiments) of fluorescence intensity of GFP-Bcl-2 (right). **$p < 0.01$, *$p < 0.05$ (one-way ANOVA). See also Supplementary Fig. 6

form of N-terminus-truncated human LRP6 (LRP6CA)[62] (a gift from Dr. A. Kikuchi); and constitutively active human Lef1 mutant (Lef1CA), in which the N-terminus β-catenin-binding region was replaced with the transactivation domain of VP16[63]. Wnt/β-catenin signalling inhibitors were as follows: human wild-type GSK-3β (a gift from Dr. A. Kikuchi); rat Axin1; dominant-negative form of C-terminus-truncated LRP6[64] (a gift from Dr. A. Kikuchi); and dominant-negative mutant of human Lef1 (Lef1DN), in which the N-terminus β-catenin binding region was truncated[65]. Other signalling regulators were as follows: baculovirus-derived p35[66] (a gift from Dr. G. Salvesen, Addgene plasmid #11808; Cambridge, MA); zebrafish E-cadherin/Cdh1 (a gift from Dr. E. Raz)[67]; E-cadherin αC mutant, in which zebrafish E-cadherin N-terminal fragment (1–794 a.a.) lacking the C-terminal β-catenin-binding domain was fused to α-catenin (constructed with reference to mouse E-cadherin αC mutant[68]; zebrafish *skilb*; dominant-negative zebrafish Smad2 mutant (Smad2 DN), in which Pro446, Ser466, and Ser468 were substituted to His, Ala and Ala, respectively[69]; dominant-negative zebrafish Smad3b mutant (Smad3 DN), in which Pro401, Ser421 and Ser423 were sub-stituted to His, Ala, and Ala, respectively[69]; constitutively active zebrafish Smad3b mutant (Smad3 CA), in which both Ser421 and Ser423 were substituted to Asp; human SOD1 (a gift from Dr. E. Fisher, Addgene #26407)[70]; zebrafish Sephs1 (transOMICS tech, Huntsville, AL); human c-Myc (a gift from W. El-Deiry, Addgene #16011)[71]; dominant-negative mutant of rat JNK2 (JNK DN), in which the MAPKK-phosphorylation sites Thr-Pro-Tyr were substituted to Ala-Pro-Phe mutants; dominant-negative mutant of *Xenopus* p38β (p38DN), in which an ATP-binding site (Lys53) was substituted to Met; the Rous sarcoma virus *src* (v-Src), which is a constitutively active Src kinase (a gift from Dr. Y. Fujita)[72]; and an oncogenic constitutively active mutant of human H-Ras (RasV12), in which Gly12 was substituted to Val (a gift from Dr. A. Yoshimura)[73]. To prepare Krt4P:mCherry, *Gal4* cDNA of the Krt4P:Gal4 plasmid, which expresses Gal4 in epithelia under the control of the keratin4 (Krt4) promoter[74], was replaced with mCherry. For mRNA synthesis, cDNAs for signalling proteins were PCR-amplified and cloned into the multi-cloning site of the pCS2p + vector. Cloned signalling proteins were as follows: human Bcl-2 (a gift from Dr. S. Korsmeyer, Addgene #8768)[75], p35, zebrafish E-cadherin-GFP fusion gene (a gift from E. Raz)[67], zebrafish *dkk1b* (a gift from Dr. M. Hibi)[76], GFP-fusion zebrafish Smad2, zebrafish Smad2 DN, zebrafish Smad3a DN, zebrafish Smad3b DN, zebrafish Skilb, human SOD1, GFP-fusion zebrafish Bcl-2 (GFP-Bcl2) and caspase activity fluorescent biosensor/VC3Ai (a gift from Dr. B. Li, Addgene plasmid #78907)[77]. Probes for in situ hybridization were as follows: *otx2*[78], *pax2a*[79], *cdx4*[80], *lef1*, *skilb*, GFP, Eluc-CP and firefly luciferase. SBE-luc (pGL4.48[luc2P/SBE/Hygro] Vector) was purchased from Promega.

**Antibodies**. Primary antibodies were as follows: anti-E-cadherin (#610181, BD Bioscience, Franklin Lakes, NJ, 1:200 dilution in immunofluorescence (IF), 1:1000 in immunoblotting); anti-αTubulin (#T6074, Sigma-Aldrich, St. Louis, MO, 1/2000); mouse anti-GFP (#A-11120, Thermo Fisher, Waltham, MA, 1/100); rabbit anti-GFP (#A-11122, Thermo Fisher, 1/500); anti-active caspase-3 (#559565, BD Bioscience, 1/300); anti-β-catenin (#C7207 Sigma-Aldrich, 1/500); anti-Sephs1 (ab96542 Abcam, 1/200); and anti-mKO2 (#M168-3M, MBL, Nagoya, Japan). Secondary antibodies were as follows: AlexaFluor488-conjugated anti-mouse IgG (#A-11029, Invitrogen, Waltham, MA, 1/300) and anti-rabbit IgG (#A-11034, Invitrogen, 1/300); AlexaFluor594-conjugated anti-mouse IgG (#A-11032, Invitrogen, 1/300) and anti-rabbit IgG (#A-11037, Invitrogen, 1/300); AlexaFluor647-conjugated anti-rabbit IgG (#4414, Cell Signaling Technology, Mountain View, CA, 1/500).

**TUNEL assay**. Embryos were fixed with 4% paraformaldehyde overnight at 4 °C and dechorionated. Embryos were rehydrated using methanol. TUNEL assay was performed with the In Situ Cell Death Detection Kit, POD (Roche) according to the manufacturer's instructions.

**Cell culture and reporter gene assay**. Neruro-2a neuroblastoma cells were grown in Dulbecco's modified Eagle's medium supplemented with 10% foetal bovine serum. Cells were transfected with the expression plasmids using Polyethylenimine MW 25000 (Polysciences, Warrington, PA). Cells were seeded into 35 mm diameter plates and then transfected with OTM:Eluc-CP reporter gene plasmids (500 ng) and the pRL-EF vector (1 ng), which expresses *Renilla* luciferase under the control of the EF-1α promoter, along with the indicated expression vectors. After 48 h, Eluc-CP and *Renilla* luciferase activities were determined using the Dual luciferase assay system (Promega). The pRL-EF vector was used to normalize the transfection efficiency of the luciferase reporters. The mean and standard deviation of duplicates from one of three independent experiments are presented.

**Southern blotting**. The tail fin of adult transgenic fish was amputated using surgical scissors and transferred to lysis buffer containing 0.1 μg/μl proteinase K. The sample was incubated overnight at 55 °C, followed by standard ethanol pre-cipitation. Purified genomic DNA samples were digested with restriction endo-nuclease. Southern blot hybridization was performed using digoxigenin-labelled probe and standard methods.

**mRNA and antisense oligo MO microinjection**. Capped mRNA was synthesized using the SP6 mMessage mMachine kit (Ambion, Austin, TX) and purified using Micro Bio-Spin columns (Bio-Rad, Hercules, CA). We injected synthesized mRNA at the one-cell stage of zebrafish embryos.

To perform knock-down experiment in zebrafish embryos, antisense oligo MOs (Gene Tools, Philomath, OR) were injected into one-cell and/or two-cell stage embryos. Standard control morpholino, 5′- CCT CTT ACC TCA GTT ACA ATT TAT A-3′; E-cadherin (*cdh1*), 5′-AAA GTC TTA CCT GAA AAA GAA AAA C-3′[81] (0.3 ng); Smad4 (*smad4a*), 5′-AAT CAT ACT CAT CCT TCA CCA TCA T-3′[82] (5 ng); APC, 5′-TAG CAT ACT CTA CCT GTG CTC TTC G-3′[83] (2.5 ng); and p53, 5′-GCG CCA TTG CTT TGC AAG AAT TG-3′[84] (5 ng) were used.

**Transplantation**. In the experiment for Supplementary Fig. 3e, the embryonic cells from Tg(hsp70l:GFP-T2A-β-catCA) embryos injected with tetramethylrhodamine-conjugated dextran (Molecular Probes, Waltham, MA) were transplanted into 3.3–3.7 hpf-stage embryos obtained by crossing heterozygous Tg(hsp70l:GFP-T2A-β-catCA) fish with wild-type fish, and then exposed to heat-shock from 4.3 hpf to 5.3 hpf. The embryos were fixed with 4% paraformaldehyde at 9 hpf, and then immunostained. In the experiments for Fig. 4f and Supplementary Fig. 4f, the embryonic cells from embryos injected with either E-cadherin (*cdh1*) antisense oligo morpholino (1 ng) or standard control morpholino and tetramethylrhodamine-conjugated dextran were transplanted into 3.3 hpf to 3.7 hpf-stage wild-type embryos. The embryos were fixed at 9 hpf and immunostained.

**Chemical inhibitors**. JNK inhibitor SP600125 (TOCRIS, Bristol, UK); p38 inhi-bitor SB203580 (TOCRIS); GSK-3β inhibitor 6-Bromoindirubin-3′-oxime (BIO) (FUJIFILM Wako, Osaka, Japan), and ROS inhibitor *N*-acetyl-L-cysteine (NAC) (Sigma-Aldrich) were used.

**Western blotting**. A total of 30 zebrafish embryos were collected and their chor-ions and yolks were removed, then embryonic tissues were lysed using the same protocol as described in Embryonic cell preparation for RNA-Seq. Cells were pel-leted at 300 g for 1 min and the supernatant discarded. Cell pellets were dissolved in lysis buffer. Cell lysates were resolved by sodium dodecyl sulphate-polyacrylamide gel electrophoresis and transferred onto polyvinylidene fluoride membrane (GE Healthcare, Buckinghamshire, UK). Membranes were immunoblotted with the primary antibodies and the bound antibodies were visualized with horseradish-peroxidase-conjugated antibodies against rabbit or mouse IgG (EMD Chemicals Inc., Darmstadt, Germany) using the ChemiLumi-One L Chemiluminescent Kit

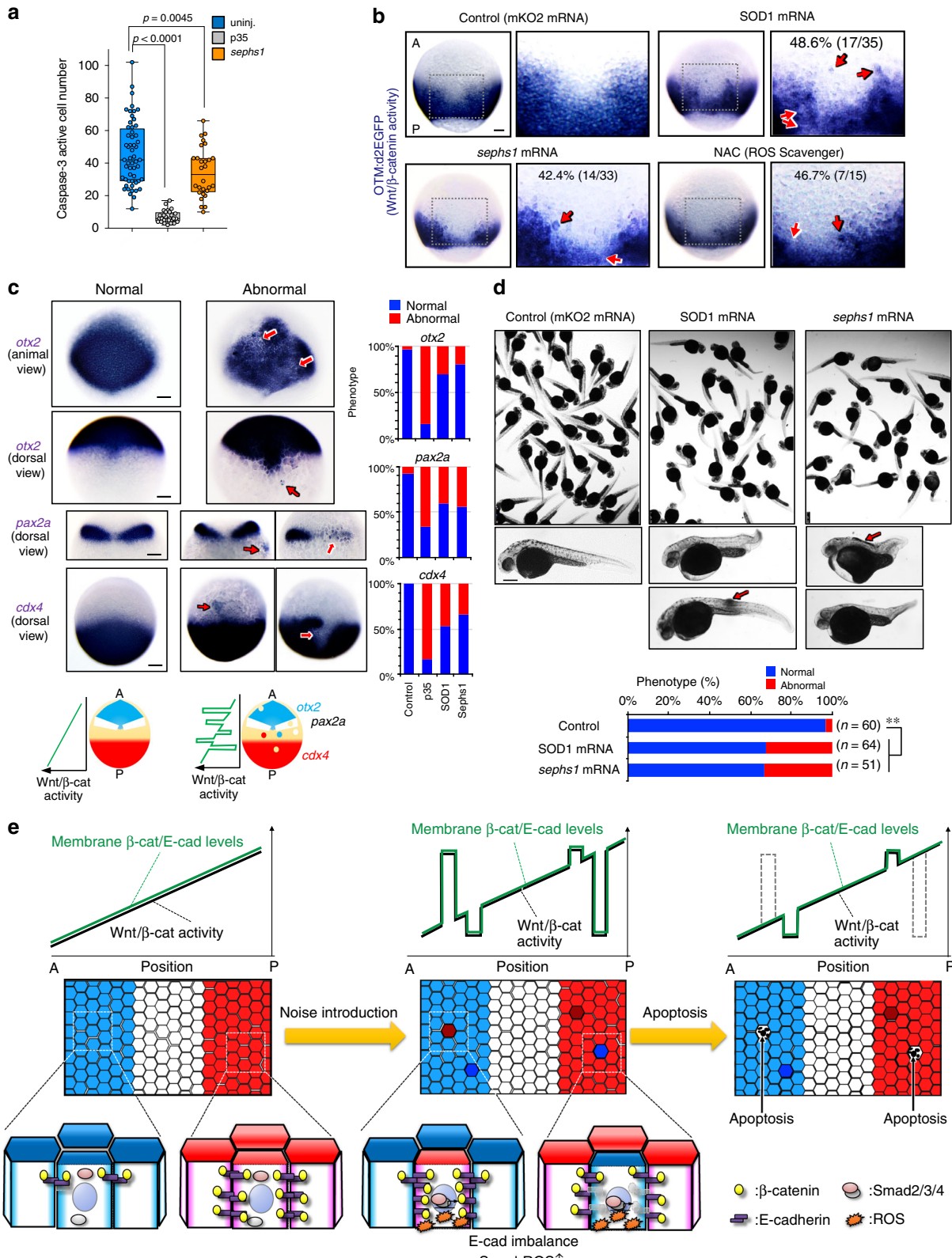

(Nacalai Tesque, Kyoto, Japan). The band intensities were measured using ImageJ. Uncropped scans of Western blots are shown in Supplementary Fig. 8.

**Quantitative reverse transcription PCR.** Total RNA from a total 25 zebrafish embryos was purified using TRIzol reagent (Invitrogen) and cDNA was synthesized with the ReverTra Ace qPCR RT Master Mix with gDNA Remover (TOYOBO, Osaka, Japan). qPCR was performed in a Mx3000P QPCR system

(Agilent Technologies, Santa Clara, CA) with THUNDERBIRD SYBR qPCR Mix (TOYOBO) and specific primers as follows: *cdh1*; 5′-TCA GTA CAG ACC TCG ACC GGC CAA-3′ (forward), 5′-AAA CAC CAG CAG AGA GTC GTA GG-3′ (reverse), *myca*; 5′-GCT TCA TCT GCG ATG ATG C-3′ (forwad), 5′-TGA TGC AGA GGT GCT CAG ATC-3′ (reverse), *mycb*; 5′-GAT GTG GTC AAC CAC AGC ATT-3′ (forwad), 5′-GGA CTC GGT CCT GGA TGA C-3′ (reverse), *dkk1b*; 5′-GCT TGG CAT GGA AGA GTT CG-3′ (forward), 5′- AGT GAC GAG CGC AGC AAA GT-3′ (reverse), and *actb1*; 5′-TGG ACT TTG AGC AGG AGA TGG

**Fig. 7** Apoptotic elimination of unfit cells is required for precise tissue patterning. **a** Sephs1 overexpression partially reduces physiologically occurring apoptosis in embryos as determined by caspase-3 immunostaining. Box plots of caspase-3-active cell number per embryo at 10 hpf stage show 75th, 50th (median) and 25th percentiles. Whiskers indicate minimum and maximum. Each dot represents one embryo ($n = 28$ embryos, two independent experiments). ****$p < 0.0001$, **$p < 0.045$ (one-way ANOVA). **b** Inhibiting ROS production distorts the Wnt/β-catenin-gradient. Whole-mount in situ hybridization of d2EGFP in Tg(OTM:d2EGFP) embryos (dorsal view) injected with $mKO2$ (control), $SOD1$, or $sephs1$ mRNA (800 pg) or treated with 100 μM NAC. Magnification of boxed area (black line) (right). Embryo percentages and numbers with similar expression patterns are shown. Red arrows: ectopic activation or inactivation areas. Scale bar, 200 μm. $p < 0.05$ for NAC treatment versus control; $p < 0.01$ for $SOD1$ or $sephs1$ mRNA versus control (Fisher's exact test). **c** Inhibition of ROS-mediated apoptosis distorts AP patterning. Panels show whole-mount in situ hybridization of $otx2$ (marker of presumptive forebrain and midbrain), $pax2a$ (marker of presumptive midbrain-hindbrain boundary), and $cdx4$ (marker of presumptive spinal cord) in embryos uninjected or injected with $mKO2$ (control), $p35$, $sephs1$, or $SOD1$ mRNA (800 pg). Scale bar, 200 μm. Bottom schematic illustration indicates expression pattern of AP tissue markers. Right graphs show percentages of embryos with normal or abnormal expression patterns. In abnormal embryos, a posterior marker ($cdx4$) and anterior markers ($pax2a$ and $otx2$) are ectopically activated in the anterior and posterior areas, respectively. **d** Overexpression of $SOD1$ or $sephs1$ mRNA induces abnormal morphogenesis. Images show 32 hpf zebrafish larvae uninjected or injected with $SOD1$ or $sephs1$ mRNA (800 pg). Red arrow indicates abnormal cell proliferation. Scale bar, 500 μm. Percentages of embryos with normal or abnormal morphology are shown. The numbers shown above the graph indicate the total numbers of embryos analysed. **$p < 0.01$ (Fisher's exact test). Note that a portion of embryos showed the anteriorization-related phenotype (e.g. short trunk and tail) or posteriorization-related phenotype (e.g. head- and eye-size reduction). A small number of embryos generated a tumour-like cell mass (arrows). **e** Schematic diagram of the Wnt/β-catenin-noise cancelling system. See also Supplementary Fig. 7

GAA-3′ (forward), 5′-AAG GTG GTC TCA TGG ATA CCG CAA-3′ (reverse). $actb1$ was used as a loading control. qPCR cycling conditions were: 95 °C for 1 min, [95 °C for 10 s, 60 °C for 30 s] (45 cycles), followed by dissociation curve analysis.

**Whole-mount immunostaining**. Embryos were fixed with 4% paraformaldehyde in phosphate-buffered saline (PBS) overnight at 4 °C. The dechorionated embryos were washed with 0.5% Triton X-100 (PBST) four times and blocked with 10% foetal bovine serum, 4% Block Ace (Megmilk Snow Brand, Tokyo, Japan), and 1% dimethylsulphoxide (DMSO) in 0.1% PBST for 1 h. The embryos were incubated with the primary antibodies overnight 4 °C, then washed and incubated with AlexaFluor-conjugated secondary antibodies overnight at 4 °C. Immunostaining for anti-8-OHdG was performed according to previously described protocol with slightly modifications[85], embryos were fixed with 50% Bounin's solution in PBS overnight at 4 °C. The dechorionated embryos were washed with 0.2% PBST six times. After rinsing with PBS, embryos were treated with RNase (100 μg/ml) in Tris buffer (10 mM Tris-HCl (pH7.5), 1 mM EDTA, and 0.4 M NaCl) at 37 °C for 1 h. After rinsing with PBS, embryos were treated with proteinase K (10 μg/mL) at room temperature for 7 min. After rinsing with PBS, DNA was denatured by treatment with 4 N HCl for 7 min at room temperature. The pH was adjusted with 50 mM Tris-HCl for 5 min at room temperature. After rinsing with 0.2% PBST and blocking, embryos were incubated with anti-8-OHdG monoclonal antibody (15A3, abcam, ab62623, 1:200) and anti-GFP antibody (Invitrogen, A-11122, 1:300) overnight 4 °C, then washed and incubated with AlexaFluor-conjugated secondary antibodies and with Hoechst33342 (Invitrogen, H3570) overnight at 4 °C. Stained embryos were visualized with the M205FA fluorescence stereo-microscope (Leica, Wetzlar, Germany), LSM700 or FV1000 or FV3000 (Olympus) confocal laser-scanning microscope, or Lightsheet Z.1 Light Sheet microscope (Zeiss). Images were prepared and analysed with ImageJ or Imaris software (Bitplane). For the quantification of membrane β-catenin and E-cadherin levels, the fluorescence intensity in two intercellular areas per cell ($n = 6$ cells) in each region was measured. Analyses were performed in duplicate.

**Whole-mount in situ hybridization**. Whole-mount in situ hybridization was performed according to standard protocol. Fluorescence in situ hybridization was performed according to the established protocol[86]. Digoxigenin- or FITC-labelled RNA antisense probes were prepared from plasmids containing $GFP$, $lef1$, $skilb$, $Eluc-CP$, $luciferase$, $otx2$, $pax2a$ and $cdx4$. Images were taken using an M205A stereo-microscope (Leica) and with an FV1000 or FV3000 confocal laser scanning microscope.

**CEL-Seq2 and data analysis**. Libraries were constructed following published protocol[87] and sequenced using the Illumina HiSeq 1500 (San Diego, CA). The reads were aligned to the zebrafish reference genome (GRCz10) using Bowtie2 software (version 2.3.1)[88]. Read counting was performed per gene with HTSeq (version 0.6.1)[89] so that duplicates in unique molecular identifiers were discarded. Using the resulting counts, genes with false discovery rates < 0.1 were extracted as differentially expressed using R library DESeq2 (version 1.10.1)[90]. Smad (Smad4)-target genes were selected using the ChIP-X database[21] (http://amp.pharm.mssm.edu/lib/chea.jsp).
  RNA-seq data have been deposited in the National Center for Biotechnology Information GEO database under the accession code GSE133526.

**Statistical analysis**. Differences between groups were examined using a two-tailed unpaired Student $t$ test, one-way ANOVA test, and Fisher's exact test in Excel

(Microsoft, Redmond, WA) or Prism7 (GraphPad Software, San Diego, CA). $p$ values < 0.05 were considered significant.

**Reporting summary**. Further information on research design is available in the Nature Research Reporting Summary linked to this article.

## Data availability
The datasets generated and/or analysed during the current study are available from the corresponding author on reasonable request. RNA-seq data have been deposited in the National Center for Biotechnology Information GEO database under the accession code GSE133526.

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

## Acknowledgements

We thank A. Kikuchi, E. Raz, A. Nagafuchi, M. Hibi, G. Salvesen, E. Fisher, W. El-Deiry, Y. Fujita, A. Yoshimura, S. Korsmeyer, M. Okada, and K. Kawakami for providing the plasmids; ZIRC and NBRP for providing transgenic zebrafish; A. Nagafuchi for helpful discussion; the Advanced Computational Scientific Program of the Research Institute for Information Technology, Kyushu University, Y. Sato in Nagoya University Live Imaging Center, Y. Kamei and T. Yabe in NIBB, J. Konno and Y. Kamihara in Olympus, NBRP zebrafish, and Ishitani lab members (Y. Sado, H. Okumura, M. Matsuo, M. Sakuma, K. Taniguchi, Y. Haraoka, M. Oginuma, and C. Mogi) for their technical support and fish maintenance. This research was supported by the Sumitomo Foundation and Takeda Foundation (T.I.), the Cooperative Research Project Program of MIB, Kyushu University (T.I. and Y.O.), AMED (JP18gm5010001) (T.I.), JSPS Research Fellowships for Young Scientists (Y.A.), JST CREST (JPMJCR16G1) (Y.O.), and a Grant-in-Aid for Scientific Research on Innovative Areas (25117720) (T.I.) (25116010, 18H04802, 18H05527, 19H05244) (Y.O.), Young Scientists (24790286,17K17942) (S.I.) (20770514) (Y.A.), and Scientific Research (B) (16H05141, 19H03412) (T.I.).

## Author contributions

Y.A. and T.I. conceived and designed the study; Y.A., S.O., H.F., S.I., R.A., J.N., T.M., N.S., Y.O. and T.I. performed the experiments; J.N. and Y.O. performed bioinformatics and data analysis; Y.A. and T.I. wrote the paper and reviewed the final draft; S.O., H.F., S.I., R.A., J.N., T.M., N.S. and Y.O. contributed to paper writing and review.

## Competing interests

The authors declare no competing interests.
