## [Peer Review File · Nature Communications]

Reviewer #1 (Remarks to the Author):

The study is a very well designed and executed attempt to establish that the embryonic cells with unusual Wnt/ β -catenin expression (in relation to the gradient) are eliminated by surrounding cells with proper Wnt/ β -catenin expression. In principle, more concrete evidence is required to establish that this process is a cell competition dependent one.

The first question which arises is the noise in the expression of the reporters (OTM:d2EGFP (Shimizu et al., 2012) and OTM:Eluc-CP reporters). The cells or pixels which are shown in Fig 1B-1D which show destabilized expression of the reporters, a through data as control must be added to establish the rate at which some cells show misfiring at the reporter levels. This data is important as the figure deals with very limited number of cells with misplaced Wnt/ β -catenin expression. With such low numbers and lack of statistics it is important to prove beyond doubt that the reporting system itself is not noisy, which in fact is to be expected.

In figure 1B-1D this reviewer would like imaging of the pixels which represent cells with irregular Wnt/ β -catenin expression. It is important to see cellular aspects of these cells with such defects in signaling.

It is shown that inhibition of apoptosis increases accumulation of cells with abnormal Wnt/ β -catenin expression. However, it is also important to show this data as accumulation of pixels as shown in 1D. In addition, it is also important to show that cells with abnormal Wnt/ β -catenin expression also show accumulation of caspases in this original setting.

In figure 2 I see a major conceptual problem with the outcome. As evident there is a lot of push to establish that the cell death is achieved via cell competition. Where cells with increased Wnt/ β -catenin expression are eliminated by surrounding cells. This has a problem as the cell competition mechanism in other embryonic studies is largely driven by MYC and it is reasonable to assume that on top of this pathway the role of MYC in Zebrafish embryo will also be important and MYC is known to be expressed in these embryonic cells. The problem is that Wnt/ β -catenin expression is known to induce MYC and that MYC overexpressing cells tend to emerge as winners in a competition scenario. Whereas in this case they will come out as losers. Which is contrary to the cell competition dogma. Authors must address this issue, either prove that this mechanism is MYC independent which is hard to believe or find a mechanism via which this anomaly can be explained.

Authors write "To identify the actual mediators, we screened the genes involved in unfit cell-killing. We collected β -catCA-expressing cells from β -catCA-mosaically introduced (Mosaic) or β -catCA ubiquitously-expressing (Ubiquitous) embryos, or cells from uninjected embryos (Uninjected) (Figure 5A)" Authors should clarify on technique used for collections of such few randomly generated cells. In my opinion, this RNA seq should have been ran on the Embryos collected in Fig 1 based on cell sorting. That would have been more direct and provided information which is physiologically more relevant. It is surprising that polarity linked genes involved in cell competition at the physiological stage of embryo development were not identified.

Since authors claim that final act of apoptosis after all the genetic events are completed is executed by increased ROS. TO this effect it is important to show if such a ROS-induced effect can be reversed by use of strong anti-oxidants.

The way the paper is written and twisted manner in which sentences are framed makes it very hard to read and understand the paper. The authors should simplify and remove twists from each and every sentence. Knowing the short coming in English due to language barrier I suggest that authors should ensure that all sentences are shortened.

Reviewer #2 (Remarks to the Author):

The Wnt morphogenetic gradient is essential for patterning the neural plate anlage in zebrafish. From posterior to anterior, a Wnt gradient is imposed on the neural plate and leads to the

induction of cell bands with hindbrain, midbrain and forebrain fate. In the work of Akieda et al. on Wnt morphogen gradients, the authors report the existence of individual cells with a beta-catenin activity which is different to the environment. These cells start to upregulation of TGFb/Smad signaling and ROS signaling. Consequently, the cells undergo apoptosis. Elimination of these cells leads to the formation of robust boundaries in the neural plate patterning. This work has been carefully done and shows convincing sets of experiments. However, I have several problems with this work which should be addressed prior acceptance.

First, the authors claim that cells - with a different beta-catenin level compared to the surrounding cells - undergo apoptosis. The authors provide evidence by overexpressing (most often) CA-beta-catenin. This tool boosts beta-catenin levels massively, and I am worrying that these cells would undergo apoptosis anyway. As a control, the authors provide evidence that beta-cat cells in a beta-cat background do not undergo apoptosis (Fig. S3E). However, the author analyzed only 4 embryos and failed to provide meaningful statistical analysis. Furthermore, it seems that - in this set of experiments - there are no apoptotic cells in beta-cat expressing embryos at all. This can hardly be. The authors should expand their analysis and provide more data showing that increase of beta-cat is not harmful to the cells of the early zf embryo. However, I would prefer if they could come up with a way to increase beta-cat level in cells within physiological levels. This is crucial as the differences of endogenous beta-cat levels are probably only minimal.

Second, the authors claim that - by blocking apoptosis by p35 mRNA or bcl2-mRNA patterning would be disrupted (Fig. 1 & Fig S2). Indeed, the authors show the appearance of some single cells expressing a different level of beta-catenin compared to the surrounding cells - outliers. However, the overall beta-catenin gradient seems quite reasonable to me. The effect is further illustrated in the schematic drawing in Fig. 7 E - highlighting single cells again. How can single cells have such a dramatic impact on patterning as claimed by the authors?

Finally, it seems that cells "know" about their position in the Wnt morphogenetic field. The authors claim that different E-cad levels are used to compare their "position" to the neighboring cells. The authors should provide a convincing line of arguments that the different E-cad level is important and that alterations influence cell survival. I could not find an experiment linking position to E-cad level to cell survival. The interpretation of experiment 2F is misleading as cells w/o E-cad detach from the tissue and undergo apoptosis - see the MZ E-cad mutant half-backed. Furthermore, the zygotic E-cad mutant does not show a patterning effect. How is this possible? A well-controlled co-culture experiment with different levels of E-cad could help to strengthen this point.

Minor comments :

Fig. 1 The authors show the gradient only by reporter expression, e.g. do you get a similar "patchy" gradient using by analyzing Wnt target gene expression for axin2 or lef1? The authors show high beta-cat activity in the Spemann organizer and in the midline (Fig. 1B, F). However, Wnts are not expressed in this region. How do the authors explain this activation?

Fig. 1B The authors claim that the cells die and disappear. Can the authors rule out that the cells change fate? Is it possible to track cells with, e.g. histone marker?

Fig. 1D Does one pixel represent one cell? The description is unclear.

Fig. 2D, E Is it possible to quantify the elimination of cells in 2D/E?

Fig. 2D/E why was the analysis carried out after 20h (D) and after 23h E?

Fig. S3C, D needs proper quantification.

Fig. 2G The authors failed to explain the rationale behind this experiment. Is apoptosis blocked in

APC MO in general? How does the number of dying cells change if beta-cat is activated in the APC MO background? It seems unlikely to me that a partial knock-down of APC leads to a nearly 100% rescue of massive activation of beta-cat activity by beta-cat-CA overexpression? I think there is still quite a difference in beta-cat levels. Furthermore, the tool APC-MO has to be adequately characterized. And, quantification is lacking.

Fig. 2H The authors claim to see a "gradient" of dying cells depending on the beta-catenin level. My question is how fast do cells undergo apoptosis? There is dramatic cell mixing in the gastrula embryo (Keller et al.). I would have envisaged that also dying cells do change position during epiboly movement. So I was wondering if the authors can track cells (e.g. high-beta-cat, low-beta-cat and surrounding cells) to support these findings.

Furthermore, it is interesting that there is a more dramatic response for apoptosis if cells have hyperactivated beta-cat levels in the lower gradient region compared to if it is an under-activate beta-cat activity in the high gradient level. Can the author explain this finding?

Fig4C/D Can this effect be titrated? Do you get a gradual change in apoptotic potential with increased MO/mRNA?

Fig. 5A, D needs proper formatting.

Fig. 6C It is essential to demonstrate where the cell clone is localized in the gastrula (AP direction) to allowing a better interpretation of the results. It seems that Axin positive cells show increased ROS signaling even though they are localized at the animal pole. How can this finding be explained?

Fig. 6B ROS positive cells are really difficult to see. Why is some of the ROS stain not related to cells where beta-cat or E-cadherin is abnormally activated? Furthermore, a quantification of the number of experiments, embryos, and cells is needed.

Fig. 7A Blocking of beta-cat mediated apoptosis in the early embryo leads to the upregulation of Smad/ROS and eventually to apoptosis of distinct cells (see Fig.1, Fig, 7b,c). However, In Fig S7 the authors show that these few cells influence the formation of the brain anlage in general (anterior & posterior area). This is in stark contrast to the points made in Fig1. How can individual misplaced cells lead to such a dramatic change in neural plate patterning?

Furthermore, the control experiment - over-expression of beta-cat - leads to no phenotype whatsoever. This is puzzling as already small amounts of beta-catenin have a strong influence on patterning - a so-called posteriorization of the neural plate. Interestingly, the authors show beta-cat embryos which display an increased otx2 expression domain - this would argue for an anteriorization. The author should support this finding by further experimental evidence and explain this finding in detail. A quantification in normal and abnormal is not helpful. The authors could measure, e.g. the area of the gene expression

Fig 7B-D, What types of abnormalities are seen?

The description of the figures is sometimes not readable (e.g. Fig. 7c and others)

Please specify the meaning of OMT.

The authors should specify details for antibodies e.g dilutions/ company and morpholinos used, eg. sequence and concentrations used

There are spelling mistakes in the figures e.g. 1.E

Reviewer #3 (Remarks to the Author):

In this work Akieda et al. have explored a new cell competition-related system that corrects noisy morphogen gradients during the development of Zebrafish. They demonstrate that unfit cells with abnormal Wnt/beta catenin signaling activity (a morphogen required to establish embryonic anterior-posterior patterning), undergo apoptosis, a process for which the communication between the unfit and neighboring fit cells, via Cadherin, is necessary. Authors propose the TGF-beta pathway as responsible for the cell apoptosis, mediated by activating Smad signaling and reactive oxygen species (ROS) production.

In general terms, work shows excellent quality, with correct experimental design and high novelty in results and conclusions. However, authors may consider the next comments before publication of the manuscript:

1. Although the involvement of TGF-beta in the apoptosis is well demonstrated, authors only focus the attention in the downstream pathways (due to the results obtained in the transcriptomic analysis that revealed changes in the levels of *skilb*) and do not mention whether TGF-beta ligands and activation of receptors are required for Smad activation. In the discussion section, they propose mechanical stress, caused by local cadherin level change, as potentially modulator of Smads. But this point is not demonstrated in the work. I recommend that authors explore, at least, the expression of *Tgfb* genes (1, 2, 3) and which are the cells that would produce them. To demonstrate the requirement (or not) of external ligands, there are excellent inhibitors of the TGF-beta Receptor I that might be used.
2. Results from ROS production (Fig. 6B) must be reinforced, due to its relevance in the molecular mechanism proposed. Data presented must be quantified and could be accompanied by analysis of oxidation in proteins and/or DNA (there are very good tools to analyze oxidized proteins or DNA by Immunofluorescence).
3. Which is the target of ROS to mediate apoptosis? Previous published data describing the axis TGF-beta/ROS as mediator of apoptosis demonstrated ROS-regulated expression of pro- and anti-apoptotic members of the Bcl-2 family, either at transcriptional or post-transcriptional level. Authors may explore it. As suggestions: *bcl-2* is a good candidate, as well as *bmf*; *mcl-1* or *bim* could be regulated at post-transcriptional level. Inhibiting ROS effects (+SOD1, for example) authors could find differences in the levels of their mRNA and/or protein.

Reviewer #4 (Remarks to the Author):

Akieda et al. describe an apoptosis/competition pathway from WNT to ROS that is activated by noisy and unequal Wnt signaling. The phenomenon is intriguing, the data of high quality and the conclusions will be of wide interest and impact, but the following points need to be addressed.

1. The beginning of the study describes the endogenous gradient and endogenously generated competitor cells but all subsequent experiments use artificial up-, down- or mis-regulation of Wnt signaling and of other components belonging to the competition pathway. It is essential to test if Cadherin, Smad, *skilb*, SOD1, *sephs1*, ROS etc are involved and regulated in the endogenous and natural context of Wnt signaling.
2. The Wnt reporter is very powerful but it is essential that beta-catenin double stainings are used

to confirm the faithful recapitulation of Wnt signaling and the existence of noise cells.

3. Does the cell in Figure 1D really die or just exit the tissue?

Authors: Akieda et al.

Title: **Cell competition corrects noisy Wnt morphogen gradients to achieve robust patterning.**

The manuscript has been revised in accordance with the comments raised by the four referees. Our responses to their comments are as follows.

Author responses to the comments of Reviewer #1

We thank Reviewer#1 for the careful and constructive review of our paper, and for the positive comment regarding the design of our study. As indicated in the responses as follows, we have taken all these comments and suggestions into account in the revised version.

1. The first question which arises is the noise in the expression of the reporters (OTM:d2EGFP) and OTM:ELuc-CP reporters. The cells or pixels which are shown in Fig 1B-1D which show destabilized expression of the reporters, a through data as control must be added to establish the rate at which some cells show misfiring at the reporter levels. This data is important as the figure deals with very limited number of cells with misplaced Wnt/ β -catenin expression. With such low numbers and lack of statistics it is important to prove beyond doubt that the reporting system itself is not noisy, which in fact is to be expected.

Response: Reviewer#1 was concerned that the noise that was visualized by the reporter system may include not only the physiological Wnt/ β -catenin noise but also reporter misfiring. As Reviewer #1 pointed out, we observed short-term transient increases and decreases in the OTM:ELuc-CP reporter activity, which are probably due to endogenous signalling fluctuation, cosmic rays, or detector noises. However, in our study, we have excluded such transient changes of the reporter activity and have focused on the noise retained for relatively long periods. In particular, in Fig 1d-e, pixels retained for ≥ 2 frames (more than 12 min) were counted as the physiological Wnt/ β -catenin noise. Pixels spontaneously showing abnormally high or low activity within a single frame (less than 12 min) were regarded as “other noise” and excluded. Thus, Fig 1d and 1e do not include the misfiring of the reporter and Fig 1e shows the dynamics of physiological Wnt/ β -catenin-noise during zebrafish AP-axis formation.

In addition, we can regard the cells with unfit OTM:d2EGFP activity as the cells maintaining abnormal Wnt/ β -catenin activity for long periods because of the EGFP property. EGFP requires approximately 25 min for its maturation (Cormack et al., Gene 1996) and the assumed half-life of d2EGFP is 2 h. Therefore, we should wait for more than 25 min to detect the increase and decrease of d2EGFP fluorescence following reporter activation and inhibition. Based on this information, we consider that the cells with unfit OTM:d2EGFP activity have maintained abnormal Wnt/ β -catenin activity for long periods and that the misfiring of the OTM:d2EGFP reporter is not detectable.

In the revised manuscript, we added the data showing the noise of endogenous Wnt/ β -catenin target gene (*lef1*) expression. *lef1* mRNA expression pattern was also noisy (Supplementary Fig 1f) and the cells with unfit *lef1* expression levels also possessed unfit OTM:d2EGFP activity (Supplementary Fig 1g), suggesting that the reporter reflects endogenous Wnt/ β -catenin target gene expression and that endogenous Wnt/ β -catenin signalling is noisy. In addition, overexpression of a ROS negative regulator Sephs1 or SOD1 induced both ectopic activation and abnormal reduction of endogenous *lef1* expression (Supplementary Fig 7c), suggesting that ROS-mediated unfit cell apoptosis eliminates the cells with abnormal endogenous Wnt/ β -catenin activity.

Furthermore, we added the data showing the number of apoptotic cells with unfit Wnt/ β -catenin activity in embryos (Supplementary Fig 2f). As Reviewer #1 predicted, the number of apoptotic cells with misplaced Wnt/ β -catenin expression was very small and varied between embryos. This likely occurs because the noise cells spontaneously appear and are actively eliminated.

2. In figure 1B-1D this reviewer would like imaging of the pixels which represent cells with irregular Wnt/ β -catenin expression. It is important to see cellular aspects of these cells with such defects in signaling. It is shown that inhibition of apoptosis increases accumulation of cells with abnormal Wnt/ β -catenin expression. However, it is also important to show this data as accumulation of pixels as shown in 1D. In addition, it is also important to show that cells with abnormal Wnt/ β -catenin expression also show accumulation of caspases in this original setting.

Response: We thank Reviewer#1 for this thoughtful comment. As Reviewer#1 suggested, we added the data showing that inhibition of apoptosis promoted the accumulation of pixels with unfit Wnt/ β -catenin activity (Supplementary Fig 2j) and that cells with unfit OTM:ELuc-CP reporter activity activated caspase-3 (Supplementary Fig 2g). [*Note that to the best of our knowledge, this is a first report of simultaneous fluorescence and bioluminescence imaging of cellular activity in a living animal. We spent considerable time towards developing this imaging system.]

In addition, we also showed that unfit Wnt/ β -catenin cells change expression/activity of the noise cancelling system regulators, including E-cadherin expression (Supplementary Fig 4d), *skilb* expression (Supplementary Fig 5f), and Smad2 activity (Supplementary Fig 5g), similar to the artificially introduced Wnt/ β -catenin-unfit cells.

3. In figure 2 I see a major conceptual problem with the outcome. As evident there is a lot of push to establish that the cell death is achieved via cell competition. Where cells with increased Wnt/ β -catenin expression are eliminated by surrounding cells. This has a problem as the cell competition mechanism in other embryonic studies is largely driven by MYC and it is reasonable to assume that on top of this pathway the role of MYC in Zebrafish embryo will also be important and MYC is known to be expressed in these embryonic cells. The problem is that Wnt/ β -catenin expression is known to induce MYC and that MYC overexpressing cells tend to emerge as winners in a competition scenario. Whereas in this case they will come out as losers. Which

is contrary to the cell competition dogma. Authors must address this issue, either prove that this mechanism is MYC independent which is hard to believe or find a mechanism via which this anomaly can be explained.

Response: We appreciate this comment. As Reviewer #1 pointed out, Myc constitutes a Wnt/ β -catenin target gene, raising the possibility that cells with high Wnt/ β -catenin activity might trigger Myc-driven cell competition. However, we concluded that Myc-driven competition and Wnt/ β -catenin-noise cancelling system mechanisms differ, based on three reasons as follows:

1) In the original version of the manuscript, we showed that Myc-driven competition was cadherin- and Smad4-independent (Supplementary Figs 4g and 5k), whereas the Wnt/ β -catenin-noise cancelling system depends on these factors.

2) Overexpression of a dominant-negative mutant of Myc (Myc-DN), which lacks its transactivation domain (Dang CV et al, Nature 1989; Sawyers CL et al, Cell 1992), dramatically blocked Myc-driven cell competition (Supplementary Fig 4h), whereas it could not block the elimination of Wnt/ β -catenin-noise cells (Supplementary Fig 4i).

3) Five hour treatment with BIO, a chemical activator of Wnt/ β -catenin signalling, activated the expression of Myc genes (*myca* and *mycb*) in zebrafish embryos (Fig x1, see below), indicating that Myc genes serve as the Wnt/ β -catenin signalling-target genes in zebrafish embryos. However, 2.5 h BIO treatment was unable to activate Myc gene expression, whereas it strongly activated expression of *dkk1b*, which is also known as a Wnt/ β -catenin target gene (Fig x1), suggesting that Myc genes are not immediate early genes in Wnt/ β -catenin signalling in zebrafish embryos. Consistent with this result, RNA-Seq data shows that heat shock-mediated forced activation of Wnt/ β -catenin signalling for 2.5 h was insufficient for the upregulation of Myc gene expression (Table x1). Notably, within 2.5–3 h after heat-shock-mediated mosaic induction of Wnt signalling regulators (e.g., β -catCA, Axin), we can detect Smad nuclear translocation, ROS production, and caspase activation in the noise cells, indicating that the Wnt/ β -catenin-noise stimulates the apoptotic elimination before affecting Myc expression. Therefore, cells with unfit Wnt/ β -catenin activity would die before activating Myc-driven cell competition.

Figure x1. Myc genes (*myca* and *mycb*) are NOT immediate early genes in Wnt/ β -catenin signalling in zebrafish embryos.

Embryos were treated with 10 μ M BIO for 2.5 or 5.0 h and then mRNA expression levels of *myca*, *mycb*, and *dkk1b* (a Wnt/ β -catenin target gene) were measured by qRT-PCR.

Table x1. *Myc* mRNA expression levels from RNA-Seq data “Mosaic (artificially introduced Wnt/ β -catenin-high cells)” v.s. “Uninjected (normal cells)”.

Gene	log2FoldChange
myca	-0.01
mycb	0.46
mycn	-0.48
mych	-0.87
mycla	0.42
myclb	0.64

4. Authors write “To identify the actual mediators, we screened the genes involved in unfit cell-killing. We collected β -catCA-expressing cells from β -catCA-mosaicly introduced (Mosaic) or β -catCA ubiquitously expressing (Ubiquitous) embryos, or cells from uninjected embryos (Uninjected) (Figure 5A)” Authors should clarify on technique used for collections of such few randomly generated cells.

Response: To clarify the method used to collect the few randomly generated cells as suggested by Reviewer#1, we added the following comments “We collected... embryos (Uninjected) by using FACS (Fig 5a)” to the Results section of the revised manuscript (page 10, line 14). We also modified Fig 5A to include a schematic of the FACS procedure and we have described the detailed methods of FACS in the revised Methods section (page 18-19).

In my opinion, this RNA seq should have been ran on the Embryos collected in Fig 1 based on cell sorting. That would have been more direct and provided information which is physiologically more relevant.

Response: We agree with Reviewer#1’s opinion and we had initially planned to collect naturally generated Wnt/ β -catenin-unfit cells using FACS. However, collecting the cells with unfit Wnt/ β -catenin activity is technically difficult because of the following reasons: 1) The number of naturally generated unfit cells is very low; 2) Because the anterior-posterior pattern-forming embryos consist of cells with a variety of Wnt/ β -catenin activities (e.g., Posterior cells have high Wnt/ β -catenin activity, whereas anterior cells have low activity), it is very difficult to collect the unfit cells with abnormally high or low Wnt/ β -catenin reporter activity using FACS. Therefore, we abandoned this approach and instead developed the experimental systems for the introduction of artificial noise cells (Fig 2a) and their FACS analysis (Fig 5a). From the experiments using these systems, we found that TGF- β -type Smad activation and *Sephs1* reduction occur in the Wnt/ β -catenin-unfit cells.

In the future, we would like to develop a new experimental system for the FACS analysis of naturally generated unfit cells by utilizing the information that was obtained in this study.

It is surprising that polarity linked genes involved in cell competition at the physiological stage of embryo development were not identified.

Response: We thank the Reviewer for this comment and note that we cannot exclude the involvement of polarity-linked genes, although our RNA-Seq data indicate that their expression remained unchanged in the Wnt/ β -catenin-noise cells (data not shown). Because it is not possible to identify the change of protein levels and post-translational modification levels using RNA-Seq, there remains the possibility that polarity-linked gene products might be post-transcriptionally regulated. In future studies, they might be identified as a mediator of the Wnt/ β -catenin-noise cancelling system.

Since authors claim that final act of apoptosis after all the genetic events are completed is executed by increased ROS. TO this effect it is important to show if such a ROS-induced effect can be reversed by use of strong anti-oxidants.

Response: In the original version of our manuscript, we showed that overexpression of anti-oxidant enzymes, SOD1 or Sephs1, blocked ROS production (visualized using CellRox Green) and caspase-3 activation in the artificially introduced Wnt/ β -catenin abnormally-high and -low noise cells (original Fig 6B and 6C / revised Fig 6b and 6d). To strengthen our model, we added new data showing that the ROS-induced effect could be reversed by overexpression of SOD1 and Sephs1. As shown in revised Fig 6c, oxidized DNA was detected in the β -catCA- or Axin-expressing noise cells. Co-expression of SOD1 or Sephs1 reduced oxidized DNA in β -catCA-expressing cells. These results reinforce our model that ROS mediates apoptosis of cells with unfit Wnt/ β -catenin activity.

The way the paper is written and twisted manner in which sentences are framed makes it very hard to read and understand the paper. The authors should simplify and remove twists from each and every sentence. Knowing the short coming in English due to language barrier I suggest that authors should ensure that all sentences are shortened.

Response: As Reviwer#1 suggested, we shortened the sentences throughout the revised manuscript to enhance clarity. In addition, the revised manuscript was edited by native English speaking editors.

Reviewer #2 (Remarks to the Author):

We thank Reviewer#2 for the careful and constructive review of our paper, and for the positive comments regarding the careful performance of this study and the merit of the experimental data. As indicated in the responses as follows, we have taken all comments and suggestions into account in the revised version.

First, the authors claim that cells - with a different beta-catenin level compared to the surrounding cells - undergo apoptosis. The authors provide evidence by overexpressing (most often) CA-beta-catenin. This tool boosts beta-catenin levels massively, and I am worrying that these cells would undergo apoptosis anyway. As a control, the authors provide evidence that beta-cat cells in a beta-cat background do not undergo apoptosis (Fig. S3E). However, the author analyzed only 4 embryos and failed to provide meaningful statistical analysis. Furthermore, it seems that - in this set of experiments - there are no apoptotic cells in beta-cat expressing embryos at all. This can hardly be. The authors should expand their analysis and provide more data showing that increase of beta-cat is not harmful to the cells of the early zf embryo.

Response: We thank the Reviewer for these comments. We concluded that β -catCA overexpression does not induce apoptosis in a cell-autonomous manner and is not harmful for the cells of the early zebrafish embryos, because ubiquitous expression of β -catCA did not activate caspase 3 in the embryos (Fig 2e). In contrast, mosaically introduced β -catCA-expressing cell underwent apoptosis (Fig 2e) and treatment of the embryos with a chemical activator of Wnt/ β -catenin signalling, BIO or APC MO, which would increase Wnt/ β -catenin signalling in whole embryos, blocked the elimination of β -catCA-expressing cells (Fig. 2f, Supplementary Fig 3d). These results suggest that a large Wnt/ β -catenin activity differential between a β -catCA-expressing cell and its neighbouring cells is required for apoptosis induction of β -catCA-expressing cells. Consistent with this model, β -catCA-expressing cells transplanted into normal but not β -catCA-expressing embryonic tissue activated caspase-3 in cells contacting host cells (Supplementary Fig 3e).

Following Reviewer#2's comment, we analysed additional transplanted embryos and revised Supplementary Fig 3e. Although Reviewer#2 appeared to think that it is strange that β -catCA-expressing cells transplanted into β -catCA-expressing embryonic tissue could not activated caspase-3 at all (Supplementary Fig 3e), we are not surprised with this result because, as described above, a large Wnt/ β -catenin activity differential between a β -catCA-expressing cell and its surrounding cells is required for the apoptosis induction of β -catCA-expressing cells.

Few cells undergo apoptosis during early embryogenesis (Supplementary Fig 2a, 2b), suggesting that embryos equip the physiological systems for killing a limited number of cells, including the Wnt/ β -catenin-noise cancelling system and others. Therefore, there remains little possibility that such physiological systems may stimulate apoptosis of the transplanted cells. Notably, a very small fraction of β -catCA-expressing cells transplanted into β -catCA-expressing embryonic tissue

underwent apoptosis (Supplementary Fig 3e). It is possible that such apoptosis might be triggered by other systems.

However, I would prefer if they could come up with a way to increase beta-cat level in cells within physiological levels. This is crucial as the differences of endogenous beta-cat levels are probably only minimal.

Response: We agree with the Reviewer that this is an important point. Therefore, we used a dominant negative mutant of GSK3 β , GSK3 β DN (original Fig 2B, 2H, S4C / revised Fig 2b, 2g, Supplementary Fig 4c). GSK3 β DN can mimic the physiological Wnt/ β -catenin activation by blocking endogenous GSK3 β -mediated phosphorylation and thereby promoting the stabilization of endogenous β -catenin. Consistent with this idea, the levels of GSK3 β DN-induced activation of the Wnt/ β -catenin reporter OTM:d2EGFP in the anterior tissue were the same as or lower than those of the physiological activation of the reporter in the posterior tissue (Fig x2, see below), indicating that GSK3 β DN can mimic the physiological Wnt/ β -catenin activation in zebrafish embryos.

We showed that GSK3 β DN-expressing cells increased E-cadherin levels (Supplementary Fig 4c) and underwent apoptosis in the Wnt/ β -catenin activity-low tissue, but not in the activity-high tissue (Fig 2b, 2g), suggesting that the Wnt/ β -catenin activation at physiological levels in an unfit area is sufficient to trigger apoptotic elimination. In addition, the rate of caspase-3 activation in GSK3 β DN-expressing cells is almost at the same level as that in β -catCA-expressing cells, suggesting that the behaviour of β -catCA-expressing cells may be similar to that of GSK3 β DN-expressing cells.

Figure x2. The levels of GSK3 β DN-induced OTM:d2EGFP activation in the anterior tissue were similar to those of the physiological activation in the posterior tissue. Optical sagittal cross-section (dorsal side) in 8.3 hpf OTM:d2EGFP-transgenic embryos mosaically introduced with cells expressing membrane mKO2 with GSK3 β DN. Left panels show fluorescence of mKO2 (magenta), OTM:d2EGFP (green), and DNA (blue). Right panels show fluorescence intensity of OTM:d2EGFP. Bottom panels show the magnified views.

Second, the authors claim that - by blocking apoptosis by p35 mRNA or bcl2-mRNA patterning would be disrupted (Fig. 1 & Fig S2). Indeed, the authors show the appearance of some single cells expressing a different level of beta-catenin

compared to the surrounding cells - outliers. However, the overall beta-catenin gradient seems quite reasonable to me. The effect is further illustrated in the schematic drawing in Fig.7 E - highlighting single cells again. How can single cells have such a dramatic impact on patterning as claimed by the authors?

Response: We appreciate this query. We consider that the single abnormal cells have a dramatic impact on patterning through the following three mechanisms:

- 1) In the early embryos, almost all cells are rapidly proliferating. Therefore, the single abnormal cells would rapidly proliferate and consequently the abnormal area would expand.
- 2) Wnt/ β -catenin signalling regulates the expression of genes encoding Wnts (e.g. Wnt8b, Mattes et al. Neural Dev. 2012) in zebrafish. The single abnormal cell might affect the Wnt/ β -catenin signalling activity in its neighbouring cells by secreting such molecules.
- 3) The noisy Wnt/ β -catenin gradient would be corrected by (i) unfit cell apoptosis, (ii) migration of unfit cells to the appropriate area, and (iii) signalling feedback systems, which are discussed in our manuscript. Communication between unfit cells and their neighbouring cells would be required for these correction systems. Possibly, a surviving abnormal cell might induce malfunction of the migration- and feedback-mediated correction system in its neighbours and consequently have a significant impact on the patterning of the surrounding tissue.

In addition, to further confirm the impact of naturally generated noise cells, we visualized expansion of the Wnt/ β -catenin signalling-noise area in the apoptosis-inhibited zebrafish embryos (revised Supplementary Fig 2j).

Finally, it seems that cells “know” about their position in the Wnt morphogenetic field. The authors claim that different E-cad levels are used to compare their “position” to the neighboring cells. The authors should provide a convincing line of arguments that the different E-cad level is important and that alterations influence cell survival. I could not find an experiment linking position to E-cad level to cell survival.

Response: We thank the Reviewer for this thoughtful comment. As Reviewer#2 suggested, we performed an experiment linking position to E-cad level to cell survival. We transplanted small numbers of E-cadherin-knockdown cells into normal embryonic tissue and evaluated the apoptosis efficiency (revised Supplementary Fig 4f). Consistent with our model, transplanted cells with abnormally low E-cadherin levels efficiently activated caspase-3 in the E-cadherin-high posterior region, indicating that a large E-cadherin level differential between abnormal and neighbouring cells is required for abnormal cell apoptosis induction.

[*Note that the transplanted E-cadherin-knockdown cells tended to locate to the anterior region. This may occur because these cells might have sensed unfitness and underwent apoptosis prior to being fixed at 9 hpf (The transplantation was performed at 3.3–3.7 hpf) and/or migrated to their fit area in which E-cadherin levels are low.]

The interpretation of experiment 2F is misleading as cells w/o E-cad detach from the tissue and undergo apoptosis - see the MZ E-cad mutant half-backed. Furthermore, the zygotic E-cad mutant does not show a patterning effect. How is this possible?

Response: As Reviewer#2 mentioned, the MZ E-cadherin mutant shows cell detachment (Kane et al. Development 2005; Shimizu et al. Mech Dev 2005), indicating that a complete loss of E-cadherin induces cell detachment. In comparison, in our study, reduction of E-cadherin levels in the E-cadherin knockdown cells was partial (original Fig S4D / revised Supplementary Fig 4e) and abnormal cell detachment was not observed in the E-cadherin knockdown embryos (data not shown). In addition, E-cadherin level reduction in artificially introduced Wnt/ β -catenin-low cells was also mild (original Fig S3B, 4C / Supplementary Fig 4b, 4c). Furthermore, the anterior embryonic tissue, in which both Wnt/ β -catenin activity and E-cadherin levels are relatively low, does not show cell detachment or undergo organized morphogenetic movement during development. Thus, cell detachment appears to be not induced by partial loss of E-cadherin.

To avoid potential confusion regarding this point, we modified the description of E-cadherin knockdown experiments by inserting the words “partial knockdown” throughout the revised manuscript.

Moreover, as Reviewer#2 pointed out, patterning defects in zebrafish E-cadherin mutants have not been reported. However, we expect that the E-cadherin mutant would have minor patterning defects, which have been overlooked. As shown in Fig 7b and 7c, inhibition of unfit cell elimination by reducing ROS activity caused minor patterning defects. E-cadherin knockdown would also induce similar minor patterning defects by blocking unfit cell elimination. In a future study, we would like to examine this issue.

A well-controlled co-culture experiment with different levels of E-cad could help to strengthen this point.

Response: We thank Reviewer#2 for this suggestion. We attempted to set up a co-culture experiment by co-culturing human SW480 cells expressing human E-cadherin and an orange fluorescent protein mKO2 (fluorescent E-cadherin-overexpressing cells) with normal SW480 cells in confluent condition and tested whether E-cadherin-overexpressing cells undergo apoptosis. However, unexpectedly, E-cadherin-overexpressing cells did not die in this experimental system (Fig. x3, see below). Possibly, the Wnt/ β -catenin-unfit cell elimination system may specifically work in tissues forming Wnt/ β -catenin-gradients and/or performing dynamic morphogenesis. Consistent with this idea, keratinocytes with abnormally high and low Wnt/ β -catenin activity survived in zebrafish larval skin, in which a Wnt/ β -catenin-gradient is not formed and dynamic morphogenesis does not occur (original Fig S3H / Supplementary Fig 3h).

Figure x3. Co-culture of E-cadherin-overexpressing SW480 cells with normal SW480 cells did not induce their apoptosis.

Fluorescence images showing fluorescence of mKO2 (red), caspase-3 activation (green), and DNA (blue) in confluent SW480 cells overexpressing mKO2 alone or in conjunction with human E-cadherin co-cultured with normal SW480 cells.

Minor comments :

Fig. 1 The authors show the gradient only by reporter expression, e.g. do you get a similar “patchy” gradient using by analyzing Wnt target gene expression for axin2 or lef1?

Response: We appreciate this thoughtful comment. As Reviewer#2 suggested, we found that the gradient of *lef1* expression pattern is also noisy and detected cells with unfit *lef1* expression levels (revised Supplementary Fig 1f).

Furthermore, inhibition of ROS-mediated unfit cell elimination by overexpressing SOD1 or Sephs1 induced both ectopic activation and abnormal reduction of *lef1* expression (revised Supplementary Fig 7c). These results reinforce our model that elimination of naturally generated Wnt/ β -catenin-unfit cells is essential to achieve smooth Wnt/ β -catenin-gradient formation.

The authors show high beta-cat activity in the Spemann organizer and in the midline (Fig. 1B, F). However, Wnts are not expressed in this region. How do the authors explain this activation?

Response: As Reviewer#2 pointed out, Wnts are not expressed in the Spemann organizer and the dorsal area (midline area) at the early epiboly stage. However, Fig. 1b and 1f show OTM:d2EGFP activity at the later epiboly stage; moreover, the area with high OTM:d2EGFP activity is the dorsal posterior area, but not the Spemann organizer. Consistent with our results, Wnt8a is expressed in the dorsal posterior area at the later epiboly stage (Kelly et al. Development 1995; Ma et al. J Neurosci 2015). In addition, another reporter (OTM:ELuc-CP) and *lef1* mRNA were also strongly expressed in the dorsal posterior area at this stage (Fig 1c, Supplementary Fig 1f), suggesting that Wnt/ β -catenin signalling is actually activated in this area.

Fig. 1B The authors claim that the cells die and disappear. Can the authors rule out that the cells change fate? Is it possible to track cells with, e.g. histone marker?

Response: To further confirm that cells with unfit Wnt/ β -catenin activity die, we visualized Wnt/ β -catenin activity and apoptosis (caspase activation) simultaneously, using OTM:ELuc-CP and VC3AI reporters, and observed that naturally generated unfit Wnt/ β -catenin cells activated caspase (revised Supplementary Fig 2g). Based on these results, we can conclude that at least some unfit cells undergo apoptosis. Although we could not exclude the possibility that other unfit cells change their fate without dying, we consider that it is not necessary to rule out these possibilities because, as we discussed in the manuscript, we consider that the unfit cell apoptosis represents one of the signalling-noise correcting systems.

Fig. 1D Does one pixel represent one cell? The description is unclear.

Response: Pixel area length is 6.5 μm , whereas zebrafish deep cell diameter is less than 10 μm , indicating that the size of one pixel is almost the same or slightly smaller than that of one embryonic cell. This information is included in the legend of Fig 1d.

Fig. 2D, E Is it possible to quantify the elimination of cells in 2D/E?

Response: It is technically difficult to quantify the eliminated cells because the artificially introduced cells locate not only in the exterior tissues (e.g. skin) but also in the interior (e.g. neural tissues) at 20 hpf. Therefore, we could not show quantitative data. Instead, we showed additional new information (the percentages of embryos showing similar phenotype and number of embryos) in the revised Fig 2d.

We have also shown the quantitative data of artificially introduced abnormal cell elimination and apoptosis at an earlier stage in the original and revised Fig 2c and Supplementary Fig 3d.

Fig.2D/E why was the analysis carried out after 20h (D) and after 23h (E)?

Response: We thank Reviewer#2 for pointing out this difference. We re-checked our experimental notebook and noted that “23h” represented a mislabelling; rather, both analyses were carried out at 20 h. In the revised manuscript, we therefore combined these Figures (revised Fig 2d).

Fig. S3C, D needs proper quantification.

Response: As Reviewer#2 suggested, we improved Supplementary Fig 3c, 3d in the revised manuscript.

Fig. 2G The authors failed to explain the rationale behind this experiment. Is apoptosis blocked in APC MO in general? How does the number of dying cells change if beta-cat is activated in the APC MO background? It seems unlikely to me that a partial knock-down of APC leads to a nearly 100% rescue of massive activation of beta-cat activity by beta-cat-CA overexpression? I think there is still quite a difference in beta-cat levels. Furthermore, the tool APC-MO has to be adequately characterized. And, quantification is lacking.

Response: The APC MO that we used in this study is well characterized and used in many studies (<https://zfin.org/action/marker/citation-list/ZDB-MRPHLNO-041206-2>). This APC MO blocks the splicing of APC transcripts (Nadauld et al. JBC 2004) and strongly increases endogenous β -catenin protein levels (Fig 5B in Eisinger et al. JBC 2007) and Wnt/ β -catenin reporter activity (Fig 1C in Phelps et al. Cell 2009) in zebrafish embryos. Therefore, we used this MO as a good tool for increasing β -catenin levels in zebrafish embryos.

As shown in Supplementary Fig 3d, inhibition of epithelial β -catCA-expressing cell elimination by APC MO was partial. This may be due to the difference between β -catCA- and APC MO-induced Wnt/ β -catenin activation strength, as Reviewer#2 pointed out. However, although the effects of APC MO were partial, APC MO significantly decreased the elimination of β -catCA-expressing cells. These results are consistent with our model that a large Wnt/ β -catenin activity differential between abnormal and neighbouring cells is required for abnormal cell apoptosis induction.

We also provided the quantitative data for showing the effects of APC MO on β -catCA-expressing cell elimination (original and revised Supplementary Fig 3d).

Fig. 2H The authors claim to see a “gradient” of dying cells depending on the beta-catenin level. My question is how fast do cells undergo apoptosis? There is dramatic cell mixing in the gastrula embryo (Keller et al.). I would have envisaged that also dying cells do change position during epiboly movement. So I was wondering if the authors can track cells (e.g. high-beta-cat, low-beta-cat and surrounding cells) to support these findings.

Response: We agree with the Reviewer that this represents an important question. Caspase activation in β -catCA-expressing cells was detected 1 h after heat-shock-mediated induction of β -cat CA, suggesting that, at least, this cell elimination system is activated within 1 h. Although we expect that the actual duration must be shorter, we were not able to simultaneously detect the movement and signalling activity of a specific cell using our current experimental systems. In future, we would like to clarify this by setting up a new imaging system.

Furthermore, it is interesting that there is a more dramatic response for apoptosis if cells have hyperactivated beta-cat levels in the lower gradient region compared to if it is an under-activate beta-cat activity in the high gradient level. Can the author explain this finding?

Response: It appears that Reviewer#2 is pointing out that the apoptosis efficiency of Wnt/ β -catenin-high unfit cells is relatively higher than that of Wnt/ β -catenin-low unfit cells (original Fig 2H / revised Fig 2g). These results suggest the possibility that a reactivity difference may exist between these cells. This may be due to the speed differential between β -catenin accumulation and degradation. Forced expression of GSK3 β DN in a Wnt/ β -catenin-low area would immediately increase endogenous β -catenin and consequent unfitness, whereas forced expression of Axin in a Wnt/ β -catenin-high area would gradually reduce endogenous β -catenin through the degradation of pre-existing abundant β -catenin.

Fig4C/D Can this effect be titrated? Do you get a gradual change in apoptotic potential with increased MO/mRNA?

Response: We appreciate this comment. We have attempted to increase or reduce the dose of MO or mRNA but were unsuccessful. The higher dose of MO or mRNA severely disturbed embryonic development and a lower dose did not have any effect on the unfit cell apoptosis. Cadherin is an important adhesion molecule for organizing tissue integrity. Therefore, the window for blocking the unfit cell apoptosis without disturbing morphogenetic movement would be very narrow. Consistent with this idea, the Wnt/ β -catenin noise-induced increase or decrease of E-cadherin levels were mild (original and revised Supplementary Fig 4c).

Fig. 5A, D needs proper formatting.

Response: As Reviewer#2 suggested, we re-formatted Fig 5a and 5d in the revised manuscript.

Fig. 6C It is essential to demonstrate where the cell clone is localized in the gastrula (AP direction) to allowing a better interpretation of the results. It seems that Axin positive cells show increased ROS signaling even though they are localized at the animal pole. How can this finding be explained?

Response: As Reviewer#2 suggested, we added the data showing that cells overexpressing Axin or E-cadherin Δ C activate ROS signalling (DNA oxidization) in the posterior tissue and that β -catCA-expressing cells also activate this signalling in the anterior tissue (revised Fig 6c).

In addition, the CellRox Green data (Fig 6b) provide the lateral view, but not the animal view, of embryos. These data suggest that Axin-expressing cells activated ROS signalling in the posterior area but not in the anterior area.

Fig. 6B ROS positive cells are really difficult to see. Why is some of the ROS stain not related to cells where beta-cat or E-cadherin is abnormally activated? Furthermore, a quantification of the number of experiments, embryos, and cells is needed.

Response: As pointed out by Reviewer#2, weak fluorescence was detected in the cells locating near Wnt/ β -catenin-abnormal cells. This may be due to the fluorescence-leakage from neighbouring abnormal cells because we used a stereomicroscope in this experiment. As described above, to strengthen our idea that ROS is produced in the unfit cells, we added the data showing the confocal images of DNA oxidization in Wnt/ β -catenin-high or -low unfit cells (revised Fig 6c), along with the quantification.

Fig. 7A Blocking of beta-cat mediated apoptosis in the early embryo leads to the upregulation of Smad/ROS and eventually to apoptosis of distinct cells (see Fig. 1, Fig, 7b,c).

Response: In contrast to the Reviewer's comment, we have not shown that inhibition of unfit cell apoptosis upregulates Smad/ROS and consequent apoptosis of distinct cells. Rather, as described in the Results section of our manuscript, we have shown that inhibition of unfit Wnt/ β -catenin cell apoptosis reduced physiologically occurring apoptosis (Fig 7a) and distorted the pattern of the Wnt/ β -catenin gradient (Fig 7b) and brain anterior-posterior pattern (Fig 7c). Reviewer#2 appears to have considered that the distortion of the Wnt/ β -catenin gradient and brain anterior-posterior patterns was caused by an increase of apoptosis, whereas we consider that the elimination-escaped unfit cells caused the distortions of these patterns.

However, In Fig S7 the authors show that these few cells influence the formation of the brain anlage in general (anterior & posterior area). This is in stark contrast to the points made in Fig1. How can individual misplaced cells lead to such a dramatic change in neural plate patterning?

Response: We thank the Reviewer for these comments. We note that the purpose and experimental condition of Supplementary Fig 7 are different from those of Fig 7. Fig 7 shows that blocking apoptosis of naturally generated unfit cells causes minor distortion of the Wnt/ β -catenin gradient and brain anterior-posterior patterns, whereas Supplementary Fig 7 shows that blocking apoptosis of artificially introduced unfit cells causes a dramatic change therein. The number of naturally generated unfit cells in zebrafish embryos is considerably lower than that of artificially introduced unfit cells. As a result, their effects are quite different.

Furthermore, the control experiment - over-expression of beta-cat - leads to no phenotype whatsoever. This is puzzling as already small amounts of beta-catenin have a strong influence on patterning - a so-called posteriorization of the neural plate. Interestingly, the authors show beta-cat embryos which display an increased otx2 expression domain - this would argue for an anteriorization. The author should support this finding by further experimental evidence and explain this finding in detail. A quantification in normal and abnormal is not helpful. The authors could measure, e.g. the area of the gene expression

Response: As Reviewer#2 pointed out, it is well known that forced activation of Wnt/ β -catenin signalling in whole embryos can induce posteriorization. However, in this study, we mosaically changed Wnt/ β -catenin signalling in embryos. Mosaically introduced Wnt/ β -catenin signalling-hyperactivated cells in the anterior tissue underwent apoptosis and then were eliminated (Fig 2). Therefore, the effects of β -catCA-expressing cell introduction were minimal (Supplementary Fig 7).

Although Reviewer#2 noted that mosaic expression of β -catCA increased the *otx2* expression domain, it actually had little effect on *otx2* expression in our study (Supplementary Fig 7). Possibly, the representative data in the original version of Supplementary Fig 7 might have been misleading. We replaced the previous version of *otx2* data with a less complicated version.

Reviewer#2 suggested that we should measure the area of gene expression. However, we have not aimed to show that abnormal cell introduction induces the simple expansion and reduction of brain marker expression area in Supplementary Fig 7. In this experiment, we attempted to show the importance of the unfit cell apoptotic elimination on precise patterning by blocking apoptosis of artificially introduced β -catCA-expressing cells in zebrafish embryos. We found that the surviving (apoptosis-blocked) β -catCA-expressing cells induced ectopic expression of a posterior marker (*cdx4*) in the anterior area and abnormal reduction of anterior markers (*pax2a* and *otx2*) in the posterior area (Supplementary Fig 7). We focused on this “ectopic change”, but not “size change” of brain marker expression. Moreover, it is technically difficult to quantify such ectopic change. Therefore, we show the percentages of embryos displaying ectopic expression patterns in Supplementary Fig 7a.

Fig 7B-D, What types of abnormalities are seen?

Response: As described in our manuscript, blocking apoptosis of naturally generated unfit cells induced both ectopic activation and abnormal reduction of the Wnt/ β -catenin signalling (Fig 7b, Supplementary Fig 7c) and brain AP markers (Fig 7c). Importantly, a posterior marker (*cdx4*) and anterior markers (*pax2a* and *otx2*) were ectopically activated in the anterior and posterior areas, respectively (Fig 7c). Inhibition of the unfit cell apoptosis also induced variable phenotypes. As shown in Fig 7d, a portion of embryos showed an anteriorization-related phenotype (e.g., short trunk and tail) or posteriorization-related phenotype (e.g., head- and eye-size reduction). A small number of embryos generated a tumour-like cell mass, which might be due to ectopic activation of Wnt/ β -catenin signalling. We described these phenotypes in the legend of revised Fig 7d (page 36).

The description of the figures is sometimes not readable (e.g. Fig. 7c and others)

Response: We apologize that the original version of the Figure legends was insufficient. We modified the figure legends to address this issue in the revised manuscript; in addition, they were edited by native English speaking editors.

Please specify the meaning of OMT.

Response: As suggested, we defined OTM (Optimal TCF Motif) when first used in the manuscript (the 1st paragraph in the Results section; page 5, lines 4-5).

The authors should specify details for antibodies e.g dilutions/ company and morpholinos used, eg. sequence and concentrations used

Response: As suggested, we added the detailed information regarding the specific reagents used in the Supplementary information.

There are spelling mistakes in the figures e.g. 1.E

Response: We apologize for the spelling mistakes, which have been corrected in the revised manuscript.

Reviewer #3 (Remarks to the Author):

We thank Reviewer#3 for the careful and constructive review of our paper, and for the positive comment regarding the excellent quality of our work, correct experimental design, and high novelty of the results and conclusions. As indicated in the following responses, we have taken all these comments and suggestions into account in the revised version of our manuscript.

1. Although the involvement of TGF-beta in the apoptosis is well demonstrated, authors only focus the attention in the downstream pathways (due to the results obtained in the transcriptomic analysis that revealed changes in the levels of skiib) and do not mention whether TGF-beta ligands and activation of receptors are required for Smad activation. In the discussion section, they propose mechanical stress, caused by local cadherin level change, as potentially modulator of Smads. But this point is not demonstrated in the work. I recommend that authors explore, at least, the expression of Tgfb genes (1, 2, 3) and which are the cells that would produce them. To demonstrate the requirement (or not) of external ligands, there are excellent inhibitors of the TGF-beta Receptor 1 that might be used.

Response: We thank the Reviewer for this thoughtful comment. As Reviewer#3 suggested, we checked the expression of TGF- β -related genes and tested the involvement of TGF- β receptors. RNA-Seq data indicate that basal expression levels of TGF- β family genes and TGF- β receptor genes in embryos at the RNA-Seq-performed stage were very low (see the values of baseMean in Table x2 and x3) and that significant activation of their expression was not detected in artificially introduced Wnt/ β -catenin-high cells (see the values of log2FoldChange in Table x2 and x3). Although Activin receptor 1 (*acvr1ba* and *acvr1l*) was relatively highly expressed, the expression was also not changed in the Wnt/ β -catenin-high cells (Table x2 and x3). In addition, treatment with both the TGF- β receptor inhibitor LY364947 and the Activin/Nodal receptor inhibitor SB431542 could not block Smad reporter activation and caspase activation in β -catCA-overexpressing cells (Supplementary Figs 5l and 5m), whereas treatment with these agents blocked endogenous TGF- β signalling-mediated Smad reporter activation at 4 hpf (Fig x4) and induced the morphological phenotypes related to inhibition of TGF- β and its related signalling (data not shown), suggesting that treatment with LY364947 and SB431542 inhibits TGF- β signalling in zebrafish and that TGF- β ligands and receptors are not involved in the unfit cell elimination.

Table x2. TGF- β ligand and receptor expression levels from RNA-Seq data “Mosaic (artificially introduced Wnt/ β -catenin-high cells)” v.s. “Uninjected (normal cells)”.

	Gene	baseMean	log2FoldChange
ligands	tgfb1a	0.06	-0.03
	tgfb2	0	
	tgfb3	0.36	-0.11
	tgfb2l	0.02	0
TGF β R1	tgfbr1a	0.02	0.01
	tgfb1b	0	
TGF β R2	tgfbr2	0	
	si:dkey-101k6.5	0	
	tgfbr3	0	

Table x3. TGF- β -related ligand and receptor expression levels from RNA-Seq data “Mosaic (artificially introduced Wnt/ β -catenin-high cells)” v.s. “Uninjected (normal cells)”.

	Gene	baseMean	log2FoldChange
Activin ligand	INHBE	0	
	CABZ01074203.1	0	
	inhbb	0.73	0.06
	inhbab	0	
Nodal ligand	ndr1	0.04	-0.06
	ndr2	0	
	spaw	0.03	0.00
Activin/Nodal Receptor 1	acvr1ba	31.42	-0.72
	ACVR1C	0	
	acvr1l	7.39	-0.42
Activin/Nodal Receptor 2	acvr2b	0.12	-0.05
	acvr2aa	1.09	-0.08
	acvr2ab	0	
	ACVR2B (1 of many)	0.15	0.04

Figure x4. Treatment with both LY364947 and SB431542 blocked endogenous TGF- β signalling-mediated Smad reporter activation at 4 hpf. Representative confocal fluorescence images show DNA (blue) and Smad reporter (SBE-luc) activity (magenta) in 4.3 hpf embryos introduced with SBE-luc, either untreated (None) or treated with both LY364947 (20 μ M) and SB431542 (50 μ M). Local activation of SBE-luc was detected in untreated embryos but not in inhibitor-treated embryos. This local activation is likely induced by endogenous TGF- β (Nodal) signalling.

We note that previous studies reported that a variety of signalling regulators including ERK, CDK, and CaMKII control Smad activity in a TGF- β -independent manner (reviewed in Pauklin and Vallier. *Development* 2015), raising the possibility that these regulators may mediate cadherin-imbalance-induced Smad activation. We will investigate and report the detailed mechanisms by which cadherin-imbalance activates Smads in our next research paper.

2. Results from ROS production (Fig. 6B) must be reinforced, due to its relevance in the molecular mechanism proposed. Data presented must be quantified and could be accompanied by analysis of oxidation in proteins and/or DNA (there are very good tools to analyze oxidized proteins or DNA by Immunofluorescence).

Response: We appreciate this helpful comment. As Reviewer#3 suggested, we added the data showing the confocal images of DNA oxidization in Wnt/ β -catenin-high or -low unfit cells (revised Fig 6c), along with quantification. In addition, we found that overexpression of anti-oxidant enzymes, SOD1 and Sephs1, reduced DNA oxidization in the unfit cells (revised Fig 6c). These results reinforce that ROS is upregulated in the unfit cells.

3. Which is the target of ROS to mediate apoptosis? Previous published data describing the axis TGF-beta/ROS as mediator of apoptosis demonstrated ROS-regulated expression of pro- and anti-apoptotic members of the Bcl-2 family, either at transcriptional or post-transcriptional level. Authors may explore it. As suggestions: bcl-2 is a good candidate, as well as bmf; mcl-1 or bim could be regulated at post-transcriptional level. Inhibiting ROS effects (+SOD1, for example) authors could find differences in the levels of their mRNA and/or protein.

Response: As suggested by Reviewer#3, we tested whether Wnt/ β -catenin-noise affects the levels of *bcl-2* family gene expression. RNA-Seq data indicates that mRNA expression levels of *bcl-2* family genes in artificially introduced noise cells were almost the same as those in control normal cells (data not shown). In contrast, levels of exogenous GFP-Bcl-2 protein in Wnt/ β -catenin-high or -low unfit cells (cells overexpressing β -catCA or Axin) were significantly lower than those in control normal cells (revised Fig 6e), whereas overexpression of SOD1 rescued the β -catCA-induced Bcl-2 reduction (revised Fig 6e). These results suggest that Wnt/ β -catenin-noise is likely to trigger apoptosis by reducing the levels of Bcl-2 protein but not *bcl-2* mRNA. Consistent with this idea, overexpression of Bcl-2 dramatically blocked the unfit cell apoptosis (original and revised Fig 1f and Supplementary Fig 5a). We described these findings in the revised manuscript (page12, lines 17-23).

Reviewer #4 (Remarks to the Author):

We thank Reviewer#4 for the careful and constructive review of our paper, and for the positive comment regarding the high quality of our data and the potential wide interest and impact of our conclusions. As indicated in the responses that follow, we have taken all these comments and suggestions into account in the revised version of the manuscript.

1. The beginning of the study describes the endogenous gradient and endogenously generated competitor cells but all subsequent experiments use artificial up-, down- or mis-regulation of Wnt signaling and of other components belonging to the competition pathway. It is essential to test if Cadherin, Smad, skilb, SOD1, sephs1, ROS etc are involved and regulated in the endogenous and natural context of Wnt signaling.

Response: We thank the Reviewer for these comments. In the original version of our manuscript, we showed that overexpression of anti-oxidant enzymes reduced naturally occurring cell death (Fig 7a) and increased abnormal cells with unfit Wnt/ β -catenin activity (Fig 7b), suggesting that ROS is involved in the elimination of cells with unfit Wnt/ β -catenin activity under physiological condition. However, it is technically difficult to show the requirement of cadherin, Smad, and skilb in the physiological Wnt/ β -catenin-noise cell elimination because overexpression or knockdown of these factors strongly affects epiboly movement and dorsal-ventral patterning. We would not be able to detect the specific defects caused by blocking of the Wnt/ β -catenin-noise cell elimination in such abnormal embryos.

However, we were able to detect that Cadherin, Smad, and *skilb* are regulated in the naturally generated unfit Wnt/ β -catenin cells and we added these data in the revised manuscript. Specifically, the change of E-cadherin and *skilb* expression levels (revised Supplementary Fig 4d, 5f) and the nuclear translocation of Smad2 (revised Supplementary Fig 5g) were detected in naturally generated unfit Wnt/ β -catenin cells. Nevertheless, because the naturally generated unfit cells are very rare, we were unable to quantify these changes sufficiently.

2. The Wnt reporter is very powerful but it is essential that beta-catenin double stainings are used to confirm the faithful recapitulation of Wnt signaling and the existence of noise cells.

Response: We appreciate this helpful recommendation. As Reviewer#4 suggested, we performed double staining of endogenous β -catenin with OTM:d2EGFP. As expected, endogenous nuclear β -catenin gradient was also noisy (revised Supplementary Fig 1h) and cells with unfit β -catenin levels possessed unfit Wnt/ β -catenin activity (revised Supplementary Fig 1i). In addition, to reinforce that our Wnt reporter reflected endogenous Wnt/ β -catenin signalling, we also showed the noise of endogenous Wnt/ β -catenin target gene (*lef1*) expression. *lef1* mRNA expression pattern was also noisy (revised Supplementary Fig 1f) and the cells with unfit *lef1* expression levels also possessed unfit OTM:d2EGFP activity (revised Supplementary Fig 1g).

3. Does the cell in Figure 1D really die or just exit the tissue?

Response: We thank the Reviewer for this query. To confirm that cells with unfit Wnt/ β -catenin activity die, we visualized Wnt/ β -catenin activity and apoptosis (caspase activation) simultaneously, using OTM:ELuc-CP and VC3AI reporters and observed that naturally generated unfit Wnt/ β -catenin cells activated caspase (revised Supplementary Fig 2g). Based on this result, we can conclude that some unfit cells die.

In our experimental system, the exit of unfit Wnt/ β -catenin cells from the tissue was not detected (data not shown). In addition, the zebrafish embryo consists of epithelial and mesenchymal cells (outer and inner cells) with mesenchymal cells being in the majority. It is difficult for mesenchymal cells to scale over the epithelial sheet to exit from embryonic tissue.

In addition, we noticed that the original description regarding Fig 1d in the Results, "Unfit cells with abnormal Wnt/ β -catenin activity gradually disappeared over time" may be misleading. Therefore, we changed this phrase to "Abnormal Wnt/ β -catenin activity gradually disappeared over time" in the revised manuscript (page 5, line 15-16).

Reviewers' comments:

Reviewer #1 (Remarks to the Author):

The authors have correctly addressed my criticisms. One last minor comment, which was brought to my attention recently: The first time that cell competition was discovered to be used during development to prevent malformations in physiological development, in the absence of artificial gene manipulation, was in Merino et al., Cell 2015. This paper should be cited and highlighted.

Reviewer #2 (Remarks to the Author):

The authors addressed most of the questions. However, there is one important question which needs to be addressed regarding the general health status of the manipulated cells. For me, it seems that the authors generate unfit cells by altering the Wnt signalling pathway which is followed by an apoptotic process. The authors need to demonstrate that these cells would survive and differentiate if they would be located close to cells with a similar Wnt activity.

Therefore, it is essential to show, for example, a rescue experiment to demonstrate that the overall fitness status of the cells is not compromised. For example, the Wnt signalling pathway could be activated by an LRP6CA construct (leading to apoptosis - see Fig. 2b). Co-expression of an antagonist (such as Axin or GSK3) should rescue the observed phenotype. Consistently, induction of apoptosis by blockage of the Wnt signalling pathway (by LRP6 DN in Fig. 2b) could be rescued by a DN-GSK3 or a low amount of beta-Catenin. Expression of an inert GFP mRNA is not a proper control as cells tolerate quite a large amount of GFP.

Fig.2 and thereafter: In general, the constructs should be checked for their suitability in this experimental context. I was wondering if ubiquitous overexpression of the same amount of mRNA of GSK3, beta-Catenin CA, LRP6 etc. alters cell viability in the embryo per se. This should not be the case if the hypothesis is correct.

In Fig2b, the statistical analysis for the "dying" CAS3+ cells should be expanded and shown in a box plot. Furthermore, the position of the clone on the AP axis should be defined. It is crucial to understand if, for example, GSK3 overexpression at the animal pole (Wnt negative) does also lead to apoptosis.

Fig4 and supplementary data: it is difficult to see the differences of E-cad localisation at the membrane. The authors should improve the quality of the pictures.

Suppl 4e: The authors, should provide proper ctrl experiments for the E-Cad Morpholino-mediated knock-down.

Fig.7. b,c: The authors show that alterations can lead to patterning effects. However, it is crucial to make the link to Wnt signalling in this Figure. According to the hypothesis of the authors, apoptotic cells/patterning effects after Wnt downregulation should be only visible in the PAX2/CDX4 positive mid/hindbrain area, and cells with high Wnt signalling should show patterning alteration only in the OTX2-positive forebrain area whereas PAX2/CDX4-area should be unaffected.

Furthermore, I think the authors need to provide the proper control experiment here. Ctrl embryos have to be injected with the same amount of mRNA – an uninjected embryo does not serve as an adequate control.

Reviewer #3 (Remarks to the Author):

Authors correctly addressed all the concerns. I congratulate the authors for the work performed.

Isabel Fabregat PHD

Reviewer #4 (Remarks to the Author):

The authors went to considerable lengths to address my and the other reviewers' comments. There is still a slight concern about endogenous signaling levels and the relatively blunt manipulations, but the paper will be of great interest to many in the field and is now ready for publication.

Authors: Akieda et al.

Title: **Cell competition corrects noisy Wnt morphogen gradients to achieve robust patterning.**

We thank the reviewers and editor for their continued efforts regarding this paper. The manuscript has been revised in accordance with the comments raised by Reviewers #1 and #2 and the Editor. Our responses to their comments are as follows. We hope that the manuscript will now be suitable for publication and look forward to your response.

Author responses to the comments of Reviewer #1

1. One last minor comment, which was brought to my attention recently: The first time that cell competition was discovered to be used during development to prevent malformations in physiological development, in the absence of artificial gene manipulation, was in Merino et al., Cell 2015. This paper should be cited and highlighted.

Response:

We agree with this comment, and as Reviewer#1 suggested, we cited this paper (Merino et al., Cell 2015) and highlighted it in the Discussion section.

Author responses to the comments of Reviewer #2

1. The authors addressed most of the questions. However, there is one important question which needs to be addressed regarding the general health status of the manipulated cells. For me, it seems that the authors generate unfit cells by altering the Wnt signalling pathway which is followed by an apoptotic

process. The authors need to demonstrate that these cells would survive and differentiate if they would be located close to cells with a similar Wnt activity. Therefore, it is essential to show, for example, a rescue experiment to demonstrate that the overall fitness status of the cells is not compromised. For example, the Wnt signalling pathway could be activated by an LRP6CA construct (leading to apoptosis - see Fig. 2b). Co-expression of an antagonist (such as Axin or GSK3) should rescue the observed phenotype. Consistently, induction of apoptosis by blockage of the Wnt signalling pathway (by LRP6 DN in Fig. 2b) could be rescued by a DN-GSK3 or a low amount of beta-Catenin. Expression of an inert GFP mRNA is not a proper control as cells tolerate quite a large amount of GFP.

Response:

Reviewer #2 is concerned that the artificial alteration of Wnt signalling activity directly activates apoptotic processes in a cell–cell communication (cell competition)-independent manner. However, our results strongly suggest that artificial activation/inhibition of Wnt signalling activity does NOT directly activate apoptotic pathways and that cell competition mediates the induction of apoptosis in Wnt signalling-abnormal cells. For example, as shown in Figs 4 and 5, caspase-3 activation in artificially introduced cells with abnormally high or low Wnt signalling activity was blocked by changing the levels of E-cad protein and Smad activity, which are not mediators of apoptosis. In addition, mosaic expression of the β -catenin Y654E mutant, which lacks cadherin-binding activity, did not induce apoptosis, whereas a mosaic increase or decrease in cytoplasmic β -catenin levels activated caspase-3 (Fig 2b, 2c, 3a, 3b). These results suggest that the mosaic activation/inhibition of Wnt signalling does not directly activate apoptosis pathways, and that E-cad-mediated cell–cell communication and consequent Smad activation mediate apoptosis induced by mosaic altered Wnt signalling. Moreover, the mosaic introduction of Lef1 CA or Lef1 DN, which induces strong Wnt signalling activation or inhibitory effects, respectively, could not induce caspase-3 activation in zebrafish embryos (Fig 3a, 3b). Furthermore, mosaic Wnt signalling activation/inhibition-induced apoptosis is context dependent [Fig 2g, Supple Fig 3i (Supple Fig 3h in

the previous version)]. These results indicate that the strong mosaic activation/inhibition of Wnt signalling is not sufficient to trigger apoptosis in zebrafish embryos. The mosaic activation of Ras or Src also did not induce apoptosis [Supple Fig 3g (Supple Fig 3f in the previous version)]. Thus, in our experimental system, artificially altering signalling activity does not directly activate apoptotic processes.

Although Reviewer#2 suggested that we confirm that the co-expression of DN-GSK3 can block apoptosis in LRP6DN-expressing cells, we think that this experiment is insufficient to test the general health status of the manipulated cells. The phenotypes of cells co-expressing both LRP6 DN and DN-GSK3 must be virtually identical to those of cells expressing DN-GSK3 because LRP6 activates β -catenin through GSK-3. In addition, we have already shown that increasing Wnt signalling activity in neighbouring cells blocks the elimination of unfit cells with abnormally high Wnt signalling activity (Fig 2f, Supple Fig 3d), suggesting that these cells are fit and survive in Wnt signalling-hyperactive tissue.

Reviewer#2 mentioned that “the authors need to demonstrate that these cells would survive and differentiate if they would be located close to cells with a similar Wnt activity”. However, this would be technically difficult. The Wnt signalling gradient is formed in a variety of tissues at various developmental stages. At the late epiboly stage, which we focused on in this study, the gradient forms along the embryonic AP-axis. In larva 1 day post-fertilization, the gradient forms along the AP axis around the midbrain–hindbrain boundary and along the DV axis of the neural tube. Therefore, cell fitness changes temporally. Artificially produced cells with high Wnt signalling activity located in the posterior brain region are fit at the epiboly stage, but some of them would become unfit and eliminated at a later stage. Therefore, it is difficult to show that artificially produced cells with fit Wnt signalling activity at the later somite stage would survive at a subsequent stage.

2. Fig.2 and thereafter: In general, the constructs should be checked for their suitability in this experimental context. I was wondering if ubiquitous overexpression of the same amount of mRNA of GSK3, beta-Catenin CA, LRP6 etc. alters cell viability in the embryo per se. This should not be the case if the hypothesis is correct.

Response:

We showed that mosaic but not ubiquitous abnormal Wnt/ β -catenin activation or inhibition in zebrafish embryos strongly activates caspase-3 (Fig 2e) and interpreted this as neighbouring normal cells are required for caspase-3 activation in the mosaic cells with abnormal Wnt/ β -catenin activity. However, Reviewer#2 seems to be concerned that levels of mRNA encoding Wnt signalling mediators, which were induced by the mosaic introduction of an expression plasmid, were much higher than those induced by ubiquitous introduction and that mosaically introduced plasmid-induced excess mRNA might cause apoptosis. As mentioned above, we think that the mosaic introduction of a plasmid does not directly induce apoptosis because cells expressing β -cat or Axin expression plasmids required changes in E-cadherin levels and Smad activation for caspase-3 activation (Fig 4, 5) and because the mosaic introduction of Lef1CA or Lef1DN expression plasmids, which induce strong activation or inhibition of Wnt signalling, respectively, did not activate caspase-3 (Fig 3a, 3b).

In addition, the experiment suggested by Reviewer#2 (overexpression of the same amount of mRNA) is almost technically impossible. It is very difficult to measure the mRNA levels of a specific Wnt signaling regulator in zebrafish embryonic cells harbouring an expression plasmid. Furthermore, it is almost impossible to introduce the same amount mRNA into all embryonic cells by injection.

3. In Fig2b, the statistical analysis for the “dying” CAS3+ cells should be expanded and shown in a box plot. Furthermore, the position of the clone on

the AP axis should be defined. It is crucial to understand if, for example, GSK3 overexpression at the animal pole (Wnt negative) does also lead to apoptosis.

Response:

As Reviewer#2 suggested, we expanded Fig2b and show this in a box plot. We also added the positional data from GSK3 β -overexpressing cells on the AP axis (revised Supple Fig 3f). As expected, GSK3 β -overexpressing cells (cells with abnormally low Wnt/ β -catenin activity) efficiently activated caspase-3 in the Wnt/ β -catenin signalling-high posterior region.

Thus, the current version of data presented in Fig 2e–g and Supple Fig 3d–f strongly suggest that a large difference in Wnt/ β -catenin activity between abnormal and neighbouring cells is required for the induction of abnormal cell apoptosis.

4. Fig4 and supplementary data: it is difficult to see the differences of E-cad localisation at the membrane. The authors should improve the quality of the pictures.

Response:

We have already repeatedly captured images of E-cad localization and have provided the best one (to take these photos, we spent half a year). However, to facilitate the readers' understanding, we have also shown the quantitative data of E-cad levels.

5. Suppl 4e: The authors, should provide proper ctrl experiments for the E-Cad Morpholino-mediated knock-down.

Response:

As described in Supple Fig 4e, we injected control MO as a control. We have already provided the proper control experiment.

6. Fig.7. b,c: The authors show that alterations can lead to patterning effects. However, it is crucial to make the link to Wnt signalling in this Figure. According to the hypothesis of the authors, apoptotic cells/patterning effects after Wnt downregulation should be only visible in the PAX2/CDX4 positive mid/hindbrain area, and cells with high Wnt signalling should show patterning alteration only in the OTX2-positive forebrain area whereas PAX2/CDX4-area should be unaffected.

Response:

As Reviewer#2 suggested, we tried to show the link between Wnt signalling-defects and patterning-defects by double in situ hybridization. As shown in Supple Fig 7d, the cells with hyperactive Wnt signalling activity induced ectopic *otx2* inhibition and *cdx4* activation in the anterior area, whereas the cells with abnormally low Wnt signalling activity induced ectopic *otx2* activation and *cdx4* inhibition in the posterior area. These results are consistent with our model that accumulated Wnt signalling in unfit cells disrupts patterning.

7. Furthermore, I think the authors need to provide the proper control experiment here. Ctrl embryos have to be injected with the same amount of mRNA – an uninjected embryo does not serve as an adequate control.

Response:

As Reviewer#2 suggested, we replaced the data of uninjected embryos with those of mKO2 mRNA-injected embryos in Figs 1f and 7bc.

REVIEWERS' COMMENTS:

Reviewer #2 (Remarks to the Author):

The authors have addressed some of the questions. However, I believe the main question remains: how fit are cells expressing several constructs to up-/down-regulate Wnt signalling in a critical phase as gastrulation. It seems most of the experiments to provide at least some evidence have been done. For example, it would be important to check if the cells - which were rescued - can survive and differentiate properly. A definitive proof can only be achieved by a clean genetic experiment which is beyond the scope of this manuscript.

Response: Reviewer #2 suggested that we determine if the cells with abnormal Wnt signalling which were rescued can survive and differentiate properly at the later stage. We agree that this is an important experiment but are afraid that it is technically difficult. Specifically, because induction of fluorescence and Wnt signalling abnormality is transient using our method of abnormal cell introduction via heat-shock-driven expression plasmids, it is difficult to track the introduced abnormal cells for an extended duration. As suggested by the Editor, we have described the potential caveats of our method for abnormal cell introduction in the Methods section.